# THE CRUCIAL ROLE OF SAMPLERS IN ONLINE DIRECT PREFERENCE OPTIMIZATION

**Ruizhe Shi**[*†]   **Runlong Zhou**[*‡]   **Simon S. Du**[§]

## ABSTRACT

Direct Preference Optimization (DPO) has emerged as a stable, scalable, and efficient solution for language model alignment. Despite its empirical success, the optimization properties, particularly the impact of samplers on its convergence rates, remain under-explored. In this paper, we provide a rigorous analysis of DPO's convergence rates with different sampling strategies under the exact gradient setting, revealing a surprising separation: uniform sampling achieves ***linear*** convergence, while our proposed online sampler achieves ***quadratic*** convergence. We further adapt the sampler to practical settings by incorporating posterior distributions and logit mixing, demonstrating improvements over previous methods. For example, it outperforms vanilla DPO by over 7.4% on Safe-RLHF dataset. Our results not only offer insights into the theoretical understanding of DPO but also pave the way for further algorithm designs. Code released at this link.

## 1 INTRODUCTION

Aligning language models (LMs) to human preferences is a critical pursuit due to its great potentials to push forward artificial intelligence (AI) development, and to enable AI to serve humanity better (Ji et al., 2023b). Reinforcement learning from human feedback (RLHF) (Ziegler et al., 2019; Bai et al., 2022) has been a widely-used approach, gaining tremendous successes in aligning LMs (OpenAI, 2024). However, the multi-stage pipeline of RLHF, including reward model training and RL tuning, is sensitive to hyperparameters and costly to train. DPO (Rafailov et al., 2023) directly combines these stages and tunes LMs in an offline way, gaining popularity due to its stablility and efficiency.

The empirical success of DPO has recently sparked a significant increase in interest for understanding its theoretical properties. Through modeling RLHF as a KL-regularized contextual bandit problem or Markov decision process, many works (Xiong et al., 2024; Xie et al., 2024; Liu et al., 2024b; Khaki et al., 2024; Song et al., 2024) obtain strong theoretical results and highlight the role of samplers in DPO. Specifically, they point out drawbacks of the offline sampler in vanilla DPO, and propose on-policy sampler or other samplers as better choices, as validated empirically (Dong et al., 2024; Guo et al., 2024; Tajwar et al., 2024).

However, these theoretical explanations are largely built upon traditional RL and analyze the impact of samplers from the view of data, namely sample complexity, thus involving some impractical assumptions, such as the access to an oracle for maximum likelihood estimation (MLE). Meanwhile, from the optimization perspective, the convergence rates of gradient descent in DPO within different sampling regimes remain an under-explored question. A particular setting of our interest is to give provable guarantees for an online sampler depending on the current policy.

**Contributions.** To fill this research gap, we focus on analyzing the crucial role of samplers in DPO, from the view of optimization. Based on our theoretical findings, we can further derive a new effective approach, demonstrating advantages in empirical experiments over previous approaches. We summarize our contributions as follows:

---

[*]Equal contribution.

[†]IIIS, Tsinghua University. Email: `srz21@mails.tsinghua.edu.cn`. Part of the work was done while Ruizhe Shi was visiting the University of Washington.

[‡]University of Washington. Email: `vectorzh@cs.washington.edu`

[§]University of Washington. Email: `ssdu@cs.washington.edu`

- **Theoretical separations.** We analyze the convergence rates of DPO with various samplers under *tabular softmax parametrization*, and demonstrate theoretical advantages brought by specific samplers. Specifically, we show a separation that our proposed samplers, `DPO-Mix-R` and `DPO-Mix-P`, achieve ***quadratic*** convergence rates, while the commonly used one, `DPO-Unif`, can only achieve ***linear*** convergence rates. Numerical simulations support our results. See Section 4.

- **Practical improvements.** We design a new sampler for practical DPO. Specifically, we employ *logit mixing* to align sampling distribution to our theory. LM alignment experiments show that under the same computation budget, our method demonstrates significant advantages over baselines. On Safe-RLHF dataset, our method exhibits an over $7.4\%$ improvement over vanilla DPO. On Iterative-Prompt dataset, our method shows a $5.4\%$ improvement over vanilla DPO. See Section 5.

- **Explainability and generalizability.** We show that our theoretical framework can explain many existing DPO variants and thus provides a new perspective on their theoretical advantages, demonstrating the generalizability of this work and its potential for designing more powerful algorithms. We study the existing DPO variants, and find that after setting a posterior distribution on the response set, vanilla DPO and on-policy DPO can both be mapped to `DPO-Unif`, while Hybrid GSHF and Online GSHF (Xiong et al., 2024) can be viewed as approximations to `DPO-Mix-P` and `DPO-Mix-R`, respectively. This further validates the soundness of our theoretical findings. See Section 5.

## 2 RELATED WORK

**Theoretical study of RLHF/DPO.** Zhu et al. (2023) formulate RLHF as contextual bandits, and prove the convergence of the maximum likelihood estimator. Xiong et al. (2024) further consider KL-regularization and show the benefits in sample complexity of online exploration in DPO. Xie et al. (2024) study the online exploration problem from the perspective of KL-regularized Markov decision processes, and show provable guarantees in sample complexity of a exploration bonus. Liu et al. (2024c) investigate the overoptimization issue, and prove a finite-sample suboptimality gap. Song et al. (2024) show a separation of coverage conditions for offline DPO and online RLHF. These works primarily focus on the perspective of data, which is widely adopted in RL literature. For Xiong et al. (2024); Xie et al. (2024), their policy update iteration is to directly solve MLE instead of doing gradient descent as ours. Song et al. (2024) focus on data coverage, and have not studied the convergence rates. In contrast, this paper analyzes DPO from the perspective of optimization, offering a complementary while more practical viewpoint.

**Variants of DPO.** There are two line of works exploring the variants of DPO. 1) *Objective function*. Ψ-PO (Azar et al., 2023) changes the reward term to alternate mappings from preference pairs. RPO (Liu et al., 2024c) adds an imitation loss to mitigate the overoptimization issue. CPO (Xu et al., 2024) removes the reference policy and adds an imitation loss to ensure that the policy does not deviate too much. SimPO (Meng et al., 2024) also removes the reference policy for efficiency, while using length normalization for better length control. 2) *Sampler*. Liu et al. (2024b); Khaki et al. (2024) utilize rejection sampling to adjust the data distribution to the theoretically-optimal policy before training. On-policy DPO (Guo et al., 2024; Tajwar et al., 2024; Ding et al., 2024) emphasize the importance of the on-policy sampler. Iterative DPO (Xiong et al., 2024; Dong et al., 2024) introduces an iterative training scheme, where an online policy is used to generate data pairs, annotated by a gold reward model, and the DPO training is subsequently applied to update the policy. XPO (Xie et al., 2024) follows the setting of iterative DPO, and adds an optimistic term to the DPO objective. In this paper, we focus on the latter direction, and only study the original objective.

**Other RLHF approaches.** There is also a line of works (Munos et al., 2023; Swamy et al., 2024; Rosset et al., 2024; Zhang et al., 2024) studying RLHF from a game-theoretic perspective. Nash-MD-PG in Munos et al. (2023) uses a geometric mixture of online policy and reference policy without specifying the mixing weight. Rosset et al. (2024) re-formulates the DPO pipeline and shows theoretical guarantees for the on-policy sampler with an MLE oracle.

**Convergence analysis of policy gradient methods.** Mei et al. (2020); Agarwal et al. (2021); Mei et al. (2023) study the convergence rates of policy gradient methods under *tabular softmax*

*parametrization*, and their results have successfully shown the practical impact of analysis with exact gradient. Motivated by them, in this paper we first study DPO with access to the exact gradient.

## 3 PRELIMINARIES

**Notations.** Let $\sigma : \mathbb{R} \to \mathbb{R}$ be the sigmoid function, where $\sigma(x) = 1/(1 + \exp(-x))$. For any set $X$, $\Delta(X)$ represents the set of probability distributions over $X$. $\mathsf{sg}\,()$ is the stopping-gradient operator. Let $\mathbb{1}_k$ be a vector with $1$ on the dimension corresponding to $k$ and $0$ on others (the dimension of this vector is implicitly defined from the context).

### 3.1 STANDARD BANDIT LEARNING

Firstly, we give basic concepts of standard bandit learning, which found the basis for RLHF.

**Multi-armed bandits and contextual bandits.** A multi-armed bandit has an arm (action) space $\mathcal{Y}$ and a reward function $r : \mathcal{Y} \to [0, 1]$. A contextual bandit has a context space $\mathcal{X}$, an arm space $\mathcal{Y}$, and a reward function $r : \mathcal{X} \times \mathcal{Y} \to [0, 1]$. In this work, the user prompt is viewed as a context, and the agent response is viewed as an arm. To simplify notations, our results are stated in *multi-armed bandits* versions. The statements and proofs can be easily extended to *contextual bandits*. Thus, we will omit the prompts (contexts) and slightly abuse the notations throughout Sections 3 and 4.

**Policies.** A policy $\pi : \mathcal{X} \to \Delta(\mathcal{Y})$ maps each context to a probability simplex over the arm space. For multi-armed bandits, a policy is instead a probability distribution over the arm space. We denote $\Pi$ as the set of policies we study. Under *tabular softmax parametrization* which is common in previous works (Rafailov et al., 2023; Azar et al., 2023; Munos et al., 2023; Swamy et al., 2024), the policy $\pi$ is parameterized by $\theta \in \mathbb{R}^{|\mathcal{Y}|}$: for any $y \in \mathcal{Y}$,

$$\pi_\theta(y) = \frac{\exp(\theta_y)}{\sum_{y' \in \mathcal{Y}} \exp(\theta_{y'})} \ .$$

The goal is to find the optimal policy maximizing the expected reward (with regularization).

### 3.2 REINFORCEMENT LEARNING FROM HUMAN FEEDBACK (RLHF)

Secondly, we introduce RLHF / preference-based reinforcement learning (PBRL) problem (Wirth et al., 2017; Christiano et al., 2017; Swamy et al., 2024) and current approaches.

**Bradley-Terry (BT) model.** Given an implicit reward oracle $r : \mathcal{X} \times \mathcal{Y} \to [0, 1]$, Bradley and Terry (1952) assume that human preference distribution $p^\star : \mathcal{X} \times \mathcal{Y} \times \mathcal{Y} \to \Delta(\{0, 1\})$ satisfies:

$$p^\star(y_1 > y_2 | x) = \sigma\left(r(x, y_1) - r(x, y_2)\right) \ .$$

This means that conditioned on prompt $x$, response $y_1$ is favored over $y_2$ with probability $p^\star(y_1 > y_2 | x)$ by human annotators.

**RLHF (Ziegler et al., 2019; Bai et al., 2022).** A human preference dataset $\mathcal{D} = \{(x^{(i)}, y_w^{(i)}, y_l^{(i)})\}_{i=1}^N$ means that in the $i^{\text{th}}$ sample, $y_w^{(i)} > y_l^{(i)}$ conditioned on $x^{(i)}$. The reward function $r : \mathcal{X} \times \mathcal{Y} \to \mathbb{R}$ is learned with parameter $\phi$ using a negative log-likelihood loss:

$$\mathcal{L}_r(\phi) = -\frac{1}{N} \sum_{i=1}^N \left[ \log \sigma\left(r_\phi(x^{(i)}, y_w^{(i)}) - r_\phi(x^{(i)}, y_l^{(i)})\right) \right] \ . \tag{1}$$

Given $\pi_1, \pi_2 \in \Pi$, $\mathbb{E}_{x \sim \rho(\mathcal{X})} \mathsf{KL}(\pi_1(\cdot|x) \| \pi_2(\cdot|x))$ where $\rho(\mathcal{X})$ is a predefined probability distribution over $\mathcal{X}$ is abbreviated as $\mathsf{KL}(\pi_1 \| \pi_2)$. Based on a reference policy $\pi_{\mathsf{ref}}$, the goal of RLHF is to maximize the obtained rewards with a KL-divergence penalty:

$$\pi^\star = \underset{\pi \in \Pi}{\operatorname{argmax}} \ \underset{x \sim \rho(\mathcal{X}), y \sim \pi(\cdot|x)}{\mathbb{E}} r_\phi(x, y) - \beta \mathsf{KL}\left(\pi \| \pi_{\mathsf{ref}}\right) \ , \tag{2}$$

where $\beta \in \mathbb{R}_+$ is the regularization coefficient. Additionally, under *tabular softmax parametrization*, we can directly write out the closed-form solution (Equation (4) in Rafailov et al. (2023)):

$$\pi^\star(y|x) = \frac{1}{Z(x)} \pi_{\mathsf{ref}}(y|x) \exp\left(\frac{1}{\beta} r_\phi(x, y)\right) \ , \ \forall x \in \mathcal{X}, y \in \mathcal{Y} \ , \tag{3}$$

where $Z(x) = \sum_{y \in \mathcal{Y}} \pi_{\mathsf{ref}}(y|x) \exp\left(\frac{1}{\beta} r_\phi(x, y)\right)$ is the partition function.

**Direct Preference Optimization (DPO, Rafailov et al. (2023)).** DPO integrates reward learning with policy learning. Given the human preference dataset $\mathcal{D} = \{(x^{(i)}, y_w^{(i)}, y_l^{(i)})\}_{i=1}^N$, the DPO policy $\pi$ is learned with parameter $\theta$ using a negative log-likelihood loss:

$$\mathcal{L}_\pi(\theta) = -\frac{1}{N} \sum_{i=1}^N \log \sigma \left( \beta \log \frac{\pi_\theta(y_w^{(i)}|x^{(i)})}{\pi_{\mathsf{ref}}(y_w^{(i)}|x^{(i)})} - \beta \log \frac{\pi_\theta(y_l^{(i)}|x^{(i)})}{\pi_{\mathsf{ref}}(y_l^{(i)}|x^{(i)})} \right) ,$$

which can be directly derived by combining Equations (1) and (3).

In this paper, we look into the role of samplers in the performance of DPO. Now we formally define DPO with samplers, from the perspective of bandit algorithms. Motivated by Mei et al. (2020); Agarwal et al. (2021); Mei et al. (2023), we first consider the scenario where we know the exact loss function and its gradient with respect to the model parameter $\theta$.

**Definition 1** (Exact DPO). *Given an action set $\mathcal{Y}$, two samplers $\pi^{\mathsf{s1}}, \pi^{\mathsf{s2}} \in \Pi$ for sampling the first and second action respectively, a human preference oracle $p^\star : \mathcal{Y} \times \mathcal{Y} \to \Delta(\{0, 1\})$, and hyperparameters $\beta, \eta \in \mathbb{R}_+$, the sampling probability and DPO loss function are defined as*

$$\pi^{\mathsf{s}}(y, y') := \mathsf{sg}\left(\pi^{\mathsf{s1}}(y)\pi^{\mathsf{s2}}(y') + \pi^{\mathsf{s1}}(y')\pi^{\mathsf{s2}}(y)\right) ,$$

$$\mathcal{L}_{\mathrm{DPO}}(\theta) := -\sum_{y,y' \in \mathcal{Y}} \pi^{\mathsf{s}}(y, y') p^\star(y > y') \log \sigma \left( \beta \log \frac{\pi_\theta(y)\pi_{\mathsf{ref}}(y')}{\pi_{\mathsf{ref}}(y)\pi_\theta(y')} \right) , \tag{4}$$

*and the parameter is updated by*

$$\theta^{(t+1)} = \theta^{(t)} - \eta\alpha(\pi^{\mathsf{s1}}, \pi^{\mathsf{s2}})\nabla_\theta \mathcal{L}_{\mathrm{DPO}}(\theta^{(t)}) , \tag{5}$$

*where $\alpha(\pi^{\mathsf{s1}}, \pi^{\mathsf{s2}})$ is a sampling coefficient determined by the samplers.*

**Remark 1.** 1) $\pi^{\mathsf{s1}}$ and $\pi^{\mathsf{s2}}$ can depend on the current parameter $\theta^{(t)}$, for example, in on-policy DPO (Guo et al., 2024; Tajwar et al., 2024). Since we stopped gradients on the sampling part, we will omit the $\theta^{(t)}$ in the future occurrences for simplicity.

2) The sampling coefficient $\alpha$ is for the purpose of comparing different sampling regimes with the same learning rate. See Appendix E for detailed explanations.

3) If the sampling regime is a mixture of ①: loss function $\mathcal{L}_1$ with sampling coefficient $\alpha_1$ and ②: loss function $\mathcal{L}_2$ with sampling coefficient $\alpha_2$, the gradient update rule follows

$$\theta^{(t+1)} = \theta^{(t)} - \eta \left( \alpha_1 \nabla_\theta \mathcal{L}_1(\theta^{(t)}) + \alpha_2 \nabla_\theta \mathcal{L}_2(\theta^{(t)}) \right) .$$

Note that ① and ② can have different sets of $\pi^{\mathsf{s1}}$ and $\pi^{\mathsf{s2}}$.

In empirical studies, we do not have access to the exact gradients. Thus, we define the scenario of empirical DPO and make mild assumptions on the gradient estimation.

**Definition 2** (Empirical DPO). *Given noise scale $\sigma \in \mathbb{R}_+$, DPO $(\sigma)$ is defined as DPO with the gradient update in Equation (5) as*

$$\theta^{(t+1)} = \theta^{(t)} - \eta G^{(t)} ,$$

*where $G_y^{(t)}$, i.e. the $y$-th entry of $G^{(t)}$, is a random variable s.t. for $\forall y \in \mathcal{Y}$ ,*

$$\frac{1}{\beta A}\left(G_y^{(t)} - \alpha(\pi^{\mathsf{s1}}, \pi^{\mathsf{s2}})\nabla_{\theta_y}\mathcal{L}(\theta^{(t)})\right) \sim \text{sub-Gaussian}(\sigma^2) .$$

**Remark 2.** If the samplers are mixed, e.g., ① and ② in Remark 1, then we assume

$$\frac{1}{\beta A}\left(G_y^{(t)} - \alpha_1 \nabla_{\theta_y}\mathcal{L}_1(\theta^{(t)}) - \alpha_2 \nabla_{\theta_y}\mathcal{L}_2(\theta^{(t)})\right) \sim \text{sub-Gaussian}(\sigma^2) .$$

The closed form solution $\pi^\star$ in Equation (3) satisfies $r(y) - r(y') - \beta \log \frac{\pi^\star(y)\pi_{\mathsf{ref}}(y')}{\pi_{\mathsf{ref}}(y)\pi^\star(y')} = 0$, which thus motivates us to study the convergence rate. With the update rule formally defined, now we ask:

> ***How fast can*** $r(y) - r(y') - \beta \log \frac{\pi_{\theta(t)}(y)\pi_{\text{ref}}(y')}{\pi_{\text{ref}}(y)\pi_{\theta(t)}(y')}$ ***converge to*** $0$ ***, for*** $\forall y, y' \in \mathcal{Y}$ ***?***

We will study the convergence rates for three sampling regimes: one sampling uniformly on the action space $\mathcal{Y}$ and two with mixtures of samplers. They are defined in Definitions 3 to 5.

**Definition 3** (Uniform sampler). `DPO-Unif` *is defined as DPO with* $\pi^{\mathsf{s1}}$, $\pi^{\mathsf{s2}}$ *as*

$$\pi^{\mathsf{s1}}(\cdot) = \pi^{\mathsf{s2}}(\cdot) = \mathsf{Uniform}(\mathcal{Y}) \ ,$$

*with* $\alpha(\pi^{\mathsf{s1}}, \pi^{\mathsf{s2}}) = 2|\mathcal{Y}|^2$.

**Definition 4** (Reward-guided mixed sampler). `DPO-Mix-R` *is defined as DPO with* $\pi^{\mathsf{s1}}$, $\pi^{\mathsf{s2}}$ *as*

$$① \left\{ \begin{array}{l} \pi^{\mathsf{s1}}(\cdot) = \mathsf{Uniform}(\mathcal{Y}) \ , \\ \pi^{\mathsf{s2}}(\cdot) = \mathsf{Uniform}(\mathcal{Y}) \ , \end{array} \right. \quad ② \left\{ \begin{array}{l} \pi^{\mathsf{s1}}(\cdot) \propto \mathsf{Uniform}(\mathcal{Y}) \cdot \exp(r(\cdot)) \ , \\ \pi^{\mathsf{s2}}(\cdot) \propto \mathsf{Uniform}(\mathcal{Y}) \cdot \exp(-r(\cdot)) \ , \end{array} \right.$$

*with* $\alpha_1 = |\mathcal{Y}|^2$, $\alpha_2 = \sum_{y,y'\in\mathcal{Y}} \exp\left(r(y) - r(y')\right)$.

**Remark 3.** When we know the reward, we intuitively want the win response distribution $\pi^{s1}$ to have a positive correlation with the reward (and vice versa for the lose response distribution $\pi^{s2}$). Definition 4 does not define a practical sampler as the ground truth reward $r(\cdot)$ is unknown, but it is important to display our idea of using a mixture of sampling pairs.

**Definition 5** (Policy-difference-guided mixed sampler). `DPO-Mix-P` *is defined as DPO with* $\pi^{\mathsf{s1}}$, $\pi^{\mathsf{s2}}$ *as*

$$① \left\{ \begin{array}{l} \pi^{\mathsf{s1}}(\cdot) = \mathsf{Uniform}(\mathcal{Y}) \ , \\ \pi^{\mathsf{s2}}(\cdot) = \mathsf{Uniform}(\mathcal{Y}) \ , \end{array} \right. \quad ② \left\{ \begin{array}{l} \pi^{\mathsf{s1}}(\cdot) \propto \mathsf{Uniform}(\mathcal{Y}) \cdot (\pi_\theta(\cdot)/\pi_{\text{ref}}(\cdot))^\beta \ , \\ \pi^{\mathsf{s2}}(\cdot) \propto \mathsf{Uniform}(\mathcal{Y}) \cdot (\pi_{\text{ref}}(\cdot)/\pi_\theta(\cdot))^\beta \ , \end{array} \right.$$

*with* $\alpha_1 = |\mathcal{Y}|^2$, $\alpha_2 = \sum_{y,y'\in\mathcal{Y}} \left(\frac{\pi_\theta(y)\pi_{\text{ref}}(y')}{\pi_{\text{ref}}(y)\pi_\theta(y')}\right)^\beta$.

**Remark 4.** When we cannot know the reward, $\beta \log \frac{\pi_\theta(y)}{\pi_{\text{ref}}(y)}$ can work as a surrogate/approximation of reward $r(y)$ (Rafailov et al., 2023). In Definition 5, ② can also be written as $\pi^{\mathsf{s1}} \propto \exp(\beta(\theta - \theta_{\text{ref}}))$, $\pi^{\mathsf{s2}} \propto \exp(\beta(\theta_{\text{ref}} - \theta))$ . $\mathsf{Uniform}(\mathcal{Y})$ in Definitions 4 and 5 is for consistency with Section 5, where we adopt a posterior distribution over $\mathcal{Y}$.

## 4 MAIN RESULTS

We show our main results on convergence rates in this section. In summary, our proposed mixed samplers can provably achieve: 1) exponentially faster convergence rates *(quadratic v.s. linear)* compared with the uniform sampler in the exact gradient setting, and 2) linear convergence rates to the noise scale when we have only unbiased estimations of the gradient. Numerical simulations corroborate these theories.

### 4.1 THEORETICAL FINDINGS

We present theories regarding convergence rates of different sampling regimes for exact DPO and empirical DPO in this subsection, along with their proof sketches. We first define important notations:

$$\Delta(y, y'; \theta) := \sigma(r(y) - r(y')) - \sigma\left(\beta \log \frac{\pi_\theta(y)\pi_{\text{ref}}(y')}{\pi_{\text{ref}}(y)\pi_\theta(y')}\right) \ ,$$

$$\delta(y, y'; \theta) := r(y) - r(y') - \beta \log \frac{\pi_\theta(y)\pi_{\text{ref}}(y')}{\pi_{\text{ref}}(y)\pi_\theta(y')} \ .$$

Then we can obtain

$$\nabla_\theta \mathcal{L}(\theta) = -\beta \sum_{y,y'} \pi^{\mathsf{s}}(y, y') \Delta(y, y'; \theta) \mathbb{1}_y \ ,$$

by plugging $p^\star(y, y') = \sigma(r(y) - r(y'))$ and $\sigma(-x) = 1 - \sigma(x)$ into the derivative of Equation (4). Hence, we can derive the iteration equation for $\delta$ following the update rule of gradient descent (5):

$$\delta(y, y'; \theta^{(t+1)}) = \delta(y, y'; \theta^{(t)})$$

$$- \eta\beta\alpha(\pi^{\mathsf{s}1},\pi^{\mathsf{s}2})\sum_{y''}\left(\pi^{\mathsf{s}}(y,y'')\Delta(y,y'';\theta^{(t)}) - \pi^{\mathsf{s}}(y',y'')\Delta(y',y'';\theta^{(t)})\right) \,. \quad (6)$$

We state the common condition for the upper bounds for simplicity:

**Condition 1.** *Given an action set $\mathcal{Y}$, it satisfies $r(y) \in [0,1]$, $\forall y \in \mathcal{Y}$. $\pi_{\theta^{(0)}}$ is initialized as $\pi_{\mathsf{ref}}$, and the regularization coefficient is $\beta \in \mathbb{R}_+$. Use the learning rate $\eta = \frac{1}{\beta^2|\mathcal{Y}|}$.*

### 4.1.1 FOR EXACT DPO

For `DPO-Unif`, we have that $\pi^{\mathsf{s}}(y,y') = 2/|\mathcal{Y}|^2$, making the coefficients of each $\Delta$ on the RHS of Equation (6) identical by absolute values. To proceed, we claim a lower bound as $\sigma'\left(\log\frac{\pi_\theta(y)\pi_{\mathsf{ref}}(y')}{\pi_{\mathsf{ref}}(y)\pi_\theta(y')}\right) \geq \sigma'_{\min}$, and use Lagrange interpolation, namely $\sigma'_{\min} \leq (\sigma(x) - \sigma(y))/(x - y) \leq 1/4$, to transform $\Delta$ into $\delta$. By carefully computing the coefficients of each $\delta$ and picking learning rate, we arrive at a linear convergence. Using this linear convergence, we can turn back to bound $\sigma'_{\min}$, completing the proof. See detailed proof in Appendix A.1.1.

**Theorem 1** (Upper bound of `DPO-Unif`). *Under Condition 1, `DPO-Unif` satisfies*

$$\left|\delta(y,y';\theta^{(T)})\right| \leq 0.588^T \,, \ \forall y,y' \in \mathcal{Y} \,,$$

*where $T \in \mathbb{N}$ is the number of iterations.*

The construction of the lower bound is based on a simple 3-armed bandit setting. We use Taylor expansion to transform $\Delta$ into $\delta$, and note that the quadratic remainders can be negligible when $\theta$ is close to the optimal point. And thus the linear transformation can only achieve linear convergence. See detailed proof in Appendix A.1.2.

**Theorem 2** (Lower bound of `DPO-Unif`). *Let $|\mathcal{Y}| = 3$, $r(y_1) = 0, r(y_2) = 1/3, r(y_3) = 1$, and $\pi_{\mathsf{ref}} = \mathsf{Uniform}(\mathcal{Y})$. For any $\beta \in \mathbb{R}_+$ and learning rate $\eta \in (0, \frac{2}{\beta^2|\mathcal{Y}|}]$, there always exists small enough $\epsilon \in \mathbb{R}_+$, for any initialization $\pi_{\theta^{(0)}}$ satisfying $\max_{y,y'\in\mathcal{Y}}\left|\delta(y,y';\theta^{(0)})\right| \leq \epsilon$ and $\min_{y,y'\in\mathcal{Y}}\left|\delta(y,y';\theta^{(0)})\right| > 0$, `DPO-Unif` satisfies*

$$\max_{y,y'\in\mathcal{Y}}\left|\delta(y,y';\theta^{(T)})\right| \geq \gamma^T \,,$$

*where $T \in \mathbb{N}$ is the number of iterations and $\gamma$ is a constant depending on $\theta^{(0)}$.*

Next we elaborate the idea of transforming $\Delta$ into $\delta$ using Taylor expansion, and show how to eliminate the linear term using appropriate samplers and learning rate. For Theorem 3, we can apply Taylor expansion at $r(y_1) - r(y_2)$ (while for Theorem 4 we apply at $\beta\log\frac{\pi_{\theta^{(t)}}(y_1)\pi_{\mathsf{ref}}(y_2)}{\pi_{\mathsf{ref}}(y_1)\pi_{\theta^{(t)}}(y_2)}$), and get

$$\Delta(y_1,y_2;\theta^{(t)}) = \sigma'(r(y_1) - r(y_2))\delta(y_1,y_2;\theta^{(t)}) + \frac{\sigma''(\xi_{\mathsf{R}})}{2}\delta(y_1,y_2;\theta^{(t)})^2 \,,$$

where $\xi_{\mathsf{R}}$ is an intermediate value. If we let $\pi^{\mathsf{s}}(y_1,y_2) \propto 1/\sigma'(r(y_1) - r(y_2))$ as in Definition 4, then

$$\pi^{\mathsf{s}}(y,y'')\Delta(y,y'';\theta^{(t)}) - \pi^{\mathsf{s}}(y',y'')\Delta(y',y'';\theta^{(t)}) = \textcolor{red}{\textbf{constant}} \cdot \delta(y,y';\theta^{(t)}) + \textcolor{blue}{\textbf{quadratic term}} \,.$$

Finally we pick an appropriate $\eta$ to eliminate the initial linear term in Equation (6) and thus establish a quadratic convergence. This observation motivates our design of samplers and proofs. The detailed proofs of Theorems 3 and 4 can be found in Appendices A.2 and A.3.

**Theorem 3** (Upper bound of `DPO-Mix-R`). *Under Condition 1, `DPO-Mix-R` satisfies*

$$\left|\delta(y,y';\theta^{(T)})\right| \leq 0.5^{2^T-1} \,, \ \forall y,y' \in \mathcal{Y} \,,$$

*where $T \in \mathbb{N}$ is the number of iterations.*

**Theorem 4** (Upper bound of `DPO-Mix-P`). *Under Condition 1, `DPO-Mix-P` satisfies*

$$\left|\delta(y,y';\theta^{(T)})\right| \leq 0.611^{2^T-1} \,, \ \forall y,y' \in \mathcal{Y} \,,$$

*where $T \in \mathbb{N}$ is the number of iterations.*

**Remark 5.** `DPO-Mix-R` is not practical as the ground truth reward oracle is inaccessible. `DPO-Mix-P` is practical as it depends on only quantities we know, at the cost of a slightly slower convergence rate.

### 4.1.2 FOR EMPIRICAL DPO

As in Definition 2, exact gradients are inaccessible in practice. Here we show the guarantees of `DPO-Mix-R` and `DPO-Mix-P` with only unbiased estimation of gradients, that they can achieve linear convergence rates to the noise scale. The basic idea is to first eliminate the linear term in expectation as we do in Section 4.1.1, and then calculate the error propagation step by step. The proofs of Theorems 5 and 6 can be found in Appendix B.

**Theorem 5.** *Under Condition 1 with the noise scale $\sigma \in (0, 1/576)$,* `DPO-Mix-R` $(\sigma)$ *satisfies*

$$\sqrt{\mathbb{E}\left[\delta(y, y'; \theta^{(T)})^2\right]} \leqslant 14\sigma \ , \ \forall y, y' \in \mathcal{Y} \ ,$$

*where $T = \left\lfloor \log \frac{1}{\sigma} \right\rfloor$ is the number of iterations.*

**Theorem 6.** *Under Condition 1 with the noise scale $\sigma \in (0, 1/576)$,* `DPO-Mix-P*` $(\sigma)$ *satisfies*

$$\sqrt{\mathbb{E}\left[\delta(y, y'; \theta^{(T)})^2\right]} \leqslant 14\sigma \ , \ \forall y, y' \in \mathcal{Y} \ ,$$

*where $T = \left\lfloor \log \frac{1}{\sigma} \right\rfloor$ is the number of iterations, and* `DPO-Mix-P*` $(\sigma)$ *is* `DPO-Mix-P` $(\sigma)$ *with a rejection sampling process: each time we get $y, y' \in \mathcal{Y}$ sampled from ②, if $\psi(y, y'; \theta^{(t)}) := \left| \beta \log \frac{\pi_{\theta^{(t)}}(y)\pi_{\text{ref}}(y')}{\pi_{\text{ref}}(y)\pi_{\theta^{(t)}}(y')} \right| > 1$, then reject this data pair with probability $1 - \frac{e+e^{-1}}{e^\psi+e^{-\psi}}$ ; and $\alpha_2$ needs to be changed to $\frac{1}{2} \sum_{y,y' \in \mathcal{Y}} \min\{e^{\psi(y,y';\theta^{(t)})} + e^{-\psi(y,y';\theta^{(t)})}, e + e^{-1}\}$ .*

**Remark 6.** The rejection process is to modify the joint sampling distribution when the policy deviates too much, such that the coefficient of the quadratic term can still be bounded. This issue also exists in `DPO-Unif` $(\sigma)$, and thus its convergence rate is not guaranteed.

Although we have not provided a theoretical lower bound on `DPO-Unif` $(\sigma)$, which we would like to leave as an open problem, we can offer an intuitive explanation. In the proof of the linear convergence of `DPO-Unif`, we need to establish a lower bound $\sigma'_{\min}$ on $\sigma' \left( \log \frac{\pi_\theta(y)\pi_{\text{ref}}(y')}{\pi_{\text{ref}}(y)\pi_\theta(y')} \right)$, after which the convergence rate becomes $2 - 8\sigma'_{\min}$. However, when faced with noisy gradients, $\left| \log \frac{\pi_\theta(y)\pi_{\text{ref}}(y')}{\pi_{\text{ref}}(y)\pi_\theta(y')} \right|$ might deviate significantly when $\eta = \frac{1}{\beta A}$, indicating the instablility.

### 4.2 NUMERICAL SIMULATIONS

We verify our theoretical findings with numerical simulations in *contextual bandits*. As shown in Figure 1, the two proposed samplers `DPO-Mix-P` and `DPO-Mix-R` show great improvements over `DPO-Unif`. The detailed configurations and more results can be found in Appendix D.1.

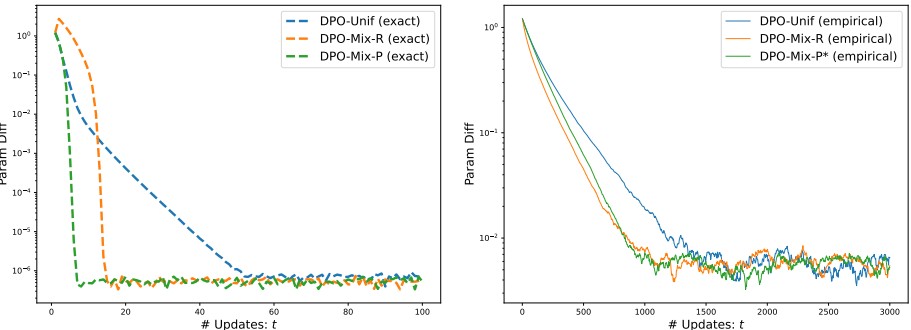

Figure 1: **Contextual bandit experiments for exact DPO and empirical DPO.** The $x$-axis is the number of gradient updates, and the $y$-axis is the total parameter difference $\sum_{y,y'} |\delta(y, y'; \theta^{(t)})|$. The left figure illustrates exact DPO, and the right figure illustrates empirical DPO. For exact DPO, the lower bound is due to the precision of floating numbers. For empirical DPO, the lower bound is due to sampling variances. The separation is clear in exact DPO, and still exists in empirical DPO.

## 5 IMPLICATIONS FOR PRACTICAL DPO

In this section, we show the implications of theoretical results in Section 4 for practical DPO design.

### 5.1 FROM THEORY TO PRACTICE

**Rethinking DPO.** We can rewrite a policy $\pi \in \Pi$ as $\pi(y|x) \propto \pi_{\mathsf{ref}}(y|x)e^{\hat{r}(x,y)/\beta}$, where $\hat{r}(x,y) \in \mathbb{R}_+$. Then the training objective of DPO can be rewritten as:

$$\sum_{y_1, y_2 \in \mathcal{Y}} \pi^{\mathsf{s}}(y_1, y_2|x) \cdot \underbrace{\left[ -\sigma\left(r(x, y_1) - r(x, y_2)\right) \log \sigma\left(\hat{r}(x, y_1) - \hat{r}(x, y_2)\right) \right]}_{\text{cross entropy loss}},$$

which is learning a reward model $\hat{r}(x, y)$ towards $r(x, y) + C(x)$, where $C(x) \in \mathbb{R}$ is a constant offset. In Section 4, we have discussed the role of samplers in this implicit reward learning stage. Here we introduce a lemma (for multi-armed bandits) to connect it with the final performance.

**Lemma 1** (Performance difference lemma). *For any $\theta$, define its value as*

$$V^\theta := \mathbb{E}_{y \sim \pi_\theta} r(y) - \beta \mathsf{KL}\left(\pi_\theta \| \pi_{\mathsf{ref}}\right),$$

*and let $V^\star$ be the value of the optimal policy $\pi^\star$ in Equation (3), then we have*

$$
\begin{aligned}
V^\star - V^\theta &= \sum_{y, y' \in \mathcal{Y}} \pi^\star(y) \pi_\theta(y') \left( r(y) - r(y') - \beta \log \frac{\pi_\theta(y) \pi_{\mathsf{ref}}(y')}{\pi_{\mathsf{ref}}(y) \pi_\theta(y')} \right) - \beta \mathsf{KL}(\pi^\star \| \pi_\theta) \\
&\leqslant \sum_{y, y' \in \mathcal{Y}} \pi^\star(y) \pi_\theta(y') \left( r(y) - r(y') - \beta \log \frac{\pi_\theta(y) \pi_{\mathsf{ref}}(y')}{\pi_{\mathsf{ref}}(y) \pi_\theta(y')} \right) \\
&= \mathbb{E}_{y \sim \pi^\star, y' \sim \pi_\theta} \delta(y, y'; \theta).
\end{aligned}
\tag{7}
$$

**Setting the posterior.** Lemma 1 indicates that $\delta(y, y'; \theta)$ contributes more to the performance when the joint probability $\pi^\star(y) \pi_\theta(y')$ is high, and thus motivates us to change the distribution over $\mathcal{Y}$ to a posterior distribution close to $\pi^\star$ or $\pi_\theta$ in practical implementation. This perspective provides an alternate explanation for Liu et al. (2024b), which uses rejection sampling to align the sampling distribution to $\pi^\star$. Considering the fact that $\pi^\star$ is usually inaccessible, we propose to let $\pi_\theta^{2\beta}$ (after normalization) be the posterior distribution. By setting the sampling temperature as $2\beta$, we can thus derive our new practical algorithm following Definition 5:

$$
① \begin{cases} \pi^{\mathsf{s}1}(\cdot|x) = \pi_\theta(\cdot|x), \\ \pi^{\mathsf{s}2}(\cdot|x) = \pi_\theta(\cdot|x), \end{cases} \quad ② \begin{cases} \pi^{\mathsf{s}1}(\cdot|x) \propto \pi_\theta^{3/2}(\cdot|x) \pi_{\mathsf{ref}}^{-1/2}(\cdot|x), \\ \pi^{\mathsf{s}2}(\cdot|x) \propto \pi_\theta^{1/2}(\cdot|x) \pi_{\mathsf{ref}}^{1/2}(\cdot|x). \end{cases}
$$

And given a reward margin $r_{\max} \in \mathbb{R}_+$, the mixing ratio can be roughly approximated as

$$① : ② = 2 : \left(\exp(r_{\max}) + \exp(-r_{\max})\right). \tag{8}$$

**Logit mixing.** The proposed samplers involve a hybridization between two policies, and a common approach to approximate hybrid distributions is *logit mixing* (Shi et al., 2024; Liu et al., 2024a). Here we show how to understand this point in a theoretically sound way. Given $\pi_1, \pi_2 \in \Pi$, $w_1, w_2 \in \mathbb{R}$, we consider a new logit as $\zeta := w_1 \zeta_1 + w_2 \zeta_2$, where $\zeta_1, \zeta_2$ represent the per-token logits of policies $\pi_1, \pi_2$, namely $\zeta_k(y_t|x, y_{<t}) = \log \pi_k(y_t|x, y_{<t})$. Note that

$$
\begin{aligned}
\underset{y \in \mathcal{Y}}{\operatorname{argmax}} \, \pi_1^{w_1}(y|x) \pi_2^{w_2}(y|x) &= \underset{y \in \mathcal{Y}}{\operatorname{argmax}} \, w_1 \log \pi_1(y|x) + w_2 \log \pi_2(y|x) \\
&= \underset{y \in \mathcal{Y}}{\operatorname{argmax}} \sum_{t=0}^{|y|} w_1 \zeta_1(y_t|x, y_{<t}) + w_2 \zeta_2(y_t|x, y_{<t}) \\
&= \underset{y \in \mathcal{Y}}{\operatorname{argmax}} \sum_{t=0}^{|y|} \zeta(y_t|x, y_{<t}).
\end{aligned}
$$

This indicates that, greedy decoding from $\pi \propto \pi_1^{w_1} \pi_2^{w_2}$ is equivalent to greedy decoding from $w_1 \zeta_1 + w_2 \zeta_2$. Thus, our proposed samplers can be implemented through mixing the logits of $\pi_{\mathsf{ref}}$ and $\pi_\theta$.

**Understanding existing approaches.** Vanilla DPO (Rafailov et al., 2023) and its online variant (Xiong et al., 2024) can be incorporated into our theoretical framework. As shown in Table 1, vanilla DPO, which assumes that pair-comparison data are sampled from $\pi_{\mathsf{ref}}$ (see Section 4 of Rafailov et al. (2023)), can be viewed as `DPO-Unif`; On-policy DPO (Guo et al., 2024; Tajwar et al., 2024) proposes to sample response pairs using $\pi_\theta$, and is thus equivalent to `DPO-Unif`; Hybrid GSHF (Option I in Xiong et al. (2024)) sets $\pi^{\mathsf{s1}} = \pi_\theta$ and $\pi^{\mathsf{s2}} = \pi_{\mathsf{ref}}$, equivalent to `DPO-Mix-P` (②); and Online GSHF (Option II in Xiong et al. (2024)) adopts the best/worst-of-$K$ response generated by $\pi_\theta$, which can be approximately viewed as generating from $\pi_\theta(\cdot)\exp(r(\cdot)/\beta)$ and $\pi_\theta(\cdot)\exp(-r(\cdot)/\beta)$, *i.e.* `DPO-Mix-R` (②). Notably, the ① part is often omitted in DPO variants, and it can be attributed to the infinitely large reward margin in the implementation (Xiong et al., 2024; Dong et al., 2024), making the mixing ratio $\to 0 : 1$ in Equation (8) (see more details in Section 4 of Rosset et al. (2024) and Appendix C).

Table 1: **Comparison with existing approaches.** We find that many baselines can be mapped to components of our proposed samplers, offering an alternative explanation for their advantages.

| Algorithm | Practical $\pi^{\mathsf{s1}}$ | Practical $\pi^{\mathsf{s2}}$ | Equivalent Sampler | Posterior |
|---|---|---|---|---|
| Vanilla DPO | $\pi_{\mathsf{ref}}$ | $\pi_{\mathsf{ref}}$ | `DPO-Unif` | $\pi_{\mathsf{ref}}^{2\beta}$ |
| On-policy DPO | $\pi_\theta$ | $\pi_\theta$ | `DPO-Unif` | $\pi_\theta^{2\beta}$ |
| Hybrid GSHF | $\pi_\theta$ | $\pi_{\mathsf{ref}}$ | `DPO-Mix-P` (②) | $\pi_\theta^\beta \pi_{\mathsf{ref}}^\beta$ |
| Online GSHF | $\pi_\theta$ (best-of-$K$) | $\pi_\theta$ (worst-of-$K$) | `DPO-Mix-R` (②) | $\pi_\theta^{2\beta}$ |
| Ours | $\pi_\theta$ $\pi_\theta^{3/2}\pi_{\mathsf{ref}}^{-1/2}$ | $\pi_\theta$ $\pi_\theta^{1/2}\pi_{\mathsf{ref}}^{1/2}$ | `DPO-Mix-P` | $\pi_\theta^{2\beta}$ |

Table 2: **Results on Safe-RLHF.** The average reward is scored by the gold reward model on train set and test set, and win-rate is against the reference model. Each algorithm is trained for 3 iterations, and in iter 3, ours shows advantages over baselines across all metrics. We repeat training each algorithm using 3 seeds, $41, 42, 43$, after iter 1, and show the mean and standard-variance.

| Algorithm | Iters | Reward (train) | Win-rate (train) | Reward (test) | Win-rate (test) |
|---|---|---|---|---|---|
| Reference | - | -2.621 | - | -2.320 | - |
| Vanilla DPO | 2 | -1.584($\pm$0.018) | 73.8($\pm$0.4)% | -1.532($\pm$0.028) | 70.4($\pm$0.9)% |
|  | 3 | -1.395($\pm$0.013) | 77.1($\pm$0.3)% | -1.372($\pm$0.001) | 73.3($\pm$0.5)% |
| On-policy DPO | 2 | -1.537($\pm$0.022) | 74.7($\pm$0.2)% | -1.490($\pm$0.023) | 71.0($\pm$0.3)% |
|  | 3 | -0.969($\pm$0.031) | 80.9($\pm$0.3)% | -0.989($\pm$0.014) | 78.3($\pm$0.2)% |
| Hybrid GSHF | 2 | -1.656($\pm$0.013) | 73.3($\pm$0.3)% | -1.580 ($\pm$0.010) | 70.2($\pm$0.4)% |
|  | 3 | -1.112($\pm$0.030) | 80.2($\pm$0.3)% | -1.039($\pm$0.111) | 78.7($\pm$0.2)% |
| Ours | 2 | -1.579($\pm$0.018) | 74.7($\pm$0.2)% | -1.490($\pm$0.022) | 71.9($\pm$0.7)% |
|  | 3 | **-0.942**($\pm$0.043) | **81.4**($\pm$0.6)% | **-0.903**($\pm$0.017) | **80.7**($\pm$0.8)% |

Table 3: **Results on Iterative-Prompt.** The average reward is scored by the gold reward model on train set and test set, and win-rate is against the reference model. Each algorithm is trained for 3 iterations, and in iter 3, ours shows advantages over baselines across all metrics. We repeat training each algorithm using 3 seeds, $41, 42, 43$, after iter 1, and show the mean and standard-variance.

| Algorithm | Iters | Reward (train) | Win-rate (train) | Reward (test) | Win-rate (test) |
|---|---|---|---|---|---|
| Reference | - | -0.665 | - | -0.639 | - |
| Vanilla DPO | 2 | 1.372($\pm$0.096) | 71.0($\pm$1.0)% | 1.459($\pm$0.065) | 71.2($\pm$0.9)% |
|  | 3 | 2.009($\pm$0.049) | 78.2($\pm$0.5)% | 2.101($\pm$0.053) | 79.0($\pm$0.9)% |
| On-policy DPO | 2 | 1.997($\pm$0.013) | 78.1($\pm$0.2)% | 2.042($\pm$0.013) | 77.5($\pm$0.4)% |
|  | 3 | 3.040($\pm$0.298) | 84.0($\pm$0.8)% | 3.132($\pm$0.315) | 83.5($\pm$0.9)% |
| Hybrid GSHF | 2 | 2.150($\pm$0.020) | 80.3($\pm$0.1)% | 2.189($\pm$0.051) | 80.3($\pm$0.7)% |
|  | 3 | 2.384($\pm$0.129) | 81.1($\pm$0.5)% | 2.490($\pm$0.160) | 82.1($\pm$1.3)% |
| Ours | 2 | 2.001($\pm$0.066) | 77.9($\pm$0.8)% | 2.047($\pm$0.017) | 77.7($\pm$0.4)% |
|  | 3 | **3.248**($\pm$0.320) | **84.8**($\pm$1.2)% | **3.321**($\pm$0.319) | **84.4**($\pm$1.7)% |

## 5.2 ALIGNMENT EXPERIMENTS

**Experiment setup.** We conduct experiments on two datasets, Safe-RLHF (Ji et al., 2023a) and Iterative-Prompt (Xiong et al., 2024; Dong et al., 2024). Our pipeline is mainly borrowed from Dong et al. (2024). For each iteration, responses are generated for a fixed set of prompts. Specifically, given prompt $x$, we generate $y_1 \sim \pi^{\mathsf{s1}}(\cdot|x)$ and $y_2 \sim \pi^{\mathsf{s2}}(\cdot|x)$ Each generated pair is annotated by a gold reward model (Dong et al., 2023) as $(r_1, r_2)$, and the corresponding loss is

$$
\mathcal{L}_{(y_1,y_2)}(\theta) = -\sigma(r_{\max} \cdot (r_1 - r_2)) \log \sigma \left( \beta \log \frac{\pi_\theta(y_1|x)\pi_{\mathsf{ref}}(y_2|x)}{\pi_{\mathsf{ref}}(y_1|x)\pi_\theta(y_2|x)} \right)
$$
$$
- \sigma(r_{\max} \cdot (r_2 - r_1)) \log \sigma \left( \beta \log \frac{\pi_\theta(y_2|x)\pi_{\mathsf{ref}}(y_1|x)}{\pi_{\mathsf{ref}}(y_2|x)\pi_\theta(y_1|x)} \right) ,
$$

where $r_{\max} \in \mathbb{R}_+$ is the reward margin. See more details in Appendix C.

**Results.** Experimental results on LM alignment are provided in Tables 2 and 3. Our setup can be viewed as a specific setting where a gold reward model is employed rather than being overfitted. We use the off-the-shelf and well-tuned reward model to simulate a real Bradley-Terry model, making the experiments cleaner and more controllable. The win-rate is simply calculated using the gold reward model. See implementations in Appendix C for details. In the final iteration, our approach shows advantages over all baselines. We also show the reward-KL curves in Figure 2, to indicate that the tuned models do not deviate too much.

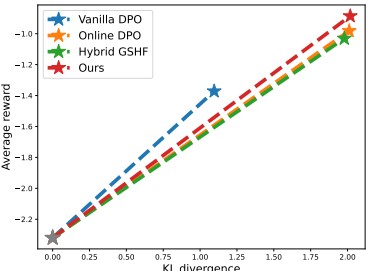 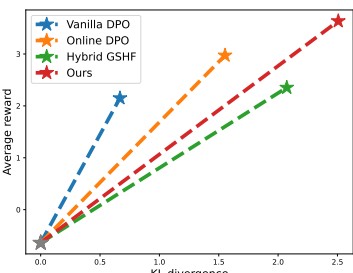

Figure 2: **The Reward-KL curves.** The left figure illustrates results on Safe-RLHF, and the right one illustrates results on Iterative-Prompt. Th KL-divergence is measured on a subset of prompts in the test set. The results indicate that the KL-divergence of trained models does not deviate much from the reference model, and our method performs best in balancing reward and KL-divergence.

**Clarification on evaluations.** It is not enough to only show the results scored by reward models, since DPO algorithm is not explicitly learning the reward rankings (Meng et al., 2024; Chen et al., 2024). Due to restricted resources, we have not evaluated on open-benchmarks (Zheng et al., 2023; Dubois et al., 2024). Our work has demonstrated the potential to train models more effectively with minimal changes to the existing DPO pipeline. We hope this will inspire the community, especially those with rich computational resources, to conduct more systematic experiments.

## 6 CONCLUSION

This paper studies the convergence rates of DPO with different samplers. We demonstrate that `DPO-Mix-R` and `DPO-Mix-P` offer quadratic convergence rates, outperforming the linear rate of `DPO-Unif`. Our theoretical findings are supported by numerical simulations and LM alignment experiments.

It is also important to acknowledge our limitations. 1) The selection of the posterior distribution is not unique, and thus many useful samplers have yet to be developed from our framework and need further experiments. 2) The convergence analysis is based on *tabular softmax parametrization*, and a future direction would be exploring more practical settings such as *log-linear parametrization* and *function approximation*. See Appendix E for a potential direction.

ACKNOWLEDGEMENT

SSD acknowledges the support of NSF IIS 2110170, NSF DMS 2134106, NSF CCF 2212261, NSF IIS 2143493, NSF IIS 2229881, Alfred P. Sloan Research Fellowship, and Schmidt Sciences AI 2050 Fellowship.

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

# Appendix

## Table of Contents

# A  PROOFS OF CONVERGENCE RATES OF EXACT DPO

Without loss of generality, we assume $\pi_{\mathsf{ref}}$ to be uniform distribution throughout this section. In the main text, we use $\mathcal{Y}$ to represent the action space and $y$ to represent an action for compatibility with other LM papers. From here, we turn back to $\mathcal{A}$ for action space, $a$ for an action, and $A$ for the size of $\mathcal{A}$ since all the proofs are conducted in bandit environments. And for notational ease, we make the following definitions:

$$\Delta(a, a'; \theta) := \sigma(r(a) - r(a')) - \sigma(\beta(\theta_a - \theta_{a'})) ,$$
$$\delta(a, a'; \theta) := r(a) - r(a') - \beta(\theta_a - \theta_{a'}) .$$

## A.1  THEOREMS 1 AND 2: LINEAR CONVERGENCE OF EXACT DPO-Unif

### A.1.1  PROOF OF UPPER BOUND

For DPO with uniform sampler on action pairs, we first claim that for any $\theta$ appearing in the optimization process,

$$\max_{a, a'}\{\beta(\theta_a - \theta_{a'})\} \leqslant R_{\max} ,$$

where $R_{\max}$ will be bounded later, and let $\sigma'_{\min} := \sigma'(R_{\max}) = \sigma(R_{\max})\sigma(-R_{\max})$ . Then we have

$$\sigma'_{\min} \leqslant \frac{\sigma(x) - \sigma(y)}{x - y} \leqslant \frac{1}{4} \text{ when } |x|, |y| \leqslant R_{\max} \text{ and } x \neq y , \tag{9}$$

$$\mathcal{L}(\theta) = -\frac{2}{A^2} \sum_{a, a'} p^\star(a > a') \log \sigma\left(\beta \log \frac{\pi_\theta(a)}{\pi_\theta(a')}\right) , \tag{10}$$

$$\nabla_\theta \mathcal{L}(\theta) = -\frac{2\beta}{A^2} \sum_{a, a'} \Delta(a, a'; \theta) \mathbb{1}_a . \tag{11}$$

Equation (11) reduces to

$$\nabla_{\theta_a} \mathcal{L}(\theta) = -\frac{2\beta}{A^2} \sum_{a'} \Delta(a, a'; \theta) .$$

Thus for any action pair $(a, a')$,

$$(\theta_a - \theta_{a'})^{(t+1)} = (\theta_a - \theta_{a'})^{(t)} + \frac{2\eta\beta\alpha(\pi^{\mathsf{s1}}, \pi^{\mathsf{s2}})}{A^2} \sum_{a''}\left(\Delta(a, a''; \theta^{(t)}) - \Delta(a', a''; \theta^{(t)})\right)$$
$$= (\theta_a - \theta_{a'})^{(t)} + 4\eta\beta \sum_{a''}\left(\Delta(a, a''; \theta^{(t)}) - \Delta(a', a''; \theta^{(t)})\right) .$$

At time $t$, sort the actions in the order that $r(a_i) - \beta\theta_{a_i}^{(t)} \leqslant r(a_{i+1}) - \beta\theta_{a_{i+1}}^{(t)}$. Then we have $\Delta(a_i, a_j; \theta^{(t)}) \geqslant 0$ if $i > j$. Note that it is possible that the order of actions at time $t+1$ is different, and in the following proof for any index $i$, $a_i$ is from the order at time $t$. Let $l < r$, then

$$\delta(a_r, a_l; \theta^{(t+1)})$$
$$= \delta(a_r, a_l; \theta^{(t)}) - 4\eta\beta^2 \sum_{i=1}^{A}\left(\Delta(a_r, a_i; \theta^{(t)}) - \Delta(a_l, a_i; \theta^{(t)})\right)$$
$$\overset{(i)}{\leqslant} \delta(a_r, a_l; \theta^{(t)}) - 4\eta\beta^2 \sum_{i=1}^{l-1}\left(\sigma'_{\min}\delta(a_r, a_i; \theta^{(t)}) - \frac{1}{4}\delta(a_l, a_i; \theta^{(t)})\right)$$
$$\quad - 4\eta\beta^2 \sum_{i=l}^{r}\left(\sigma'_{\min}\delta(a_r, a_i; \theta^{(t)}) - \sigma'_{\min}\delta(a_l, a_i; \theta^{(t)})\right) - 4\eta\beta^2 \sum_{i=r+1}^{A}\left(\frac{1}{4}\delta(a_r, a_i; \theta^{(t)}) - \sigma'_{\min}\delta(a_l, a_i; \theta^{(t)})\right)$$
$$= \delta(a_r, a_l; \theta^{(t)}) - 4\eta\beta^2 \left[\sigma'_{\min}(l - 1)\delta(a_r, a_l; \theta^{(t)}) - \left(\frac{1}{4} - \sigma'_{\min}\right)\sum_{i=1}^{l-1}\delta(a_l, a_i; \theta^{(t)})\right]$$

$$- 4\eta\beta^2\sigma'_{\min}(r - l + 1)\delta(a_r, a_l; \theta^{(t)}) - 4\eta\beta^2\left[\sigma'_{\min}(A - r)\delta(a_r, a_l; \theta^{(t)}) - \left(\frac{1}{4} - \sigma'_{\min}\right)\sum_{i=r+1}^{A}\delta(a_i, a_r; \theta^{(t)})\right]$$

$$= \left(1 - 4\eta\beta^2 A\sigma'_{\min}\right)\delta(a_r, a_l; \theta^{(t)}) + 4\eta\beta^2\left(\frac{1}{4} - \sigma'_{\min}\right)\left(\sum_{i=1}^{l-1}\delta(a_l, a_i; \theta^{(t)}) + \sum_{i=r+1}^{A}\delta(a_i, a_r; \theta^{(t)})\right),$$

where (i) is by using Equation (9) for different cases of $x$ and $y$ and whether $x - y > 0$. Similarly, for the lower bound:

$$- \delta(a_r, a_l; \theta^{(t+1)})$$

$$= 4\eta\beta^2\sum_{i=1}^{A}\left(\Delta(a_r, a_i; \theta^{(t)}) - \Delta(a_l, a_i; \theta^{(t)})\right) - \delta(a_r, a_l; \theta^{(t)})$$

$$\leqslant 4\eta\beta^2\sum_{i=1}^{l-1}\left(\frac{1}{4}\delta(a_r, a_i; \theta^{(t)}) - \sigma'_{\min}\delta(a_l, a_i; \theta^{(t)})\right) + 4\eta\beta^2\sum_{i=l}^{r}\left(\frac{1}{4}\delta(a_r, a_i; \theta^{(t)}) - \frac{1}{4}\delta(a_l, a_i; \theta^{(t)})\right)$$

$$+ 4\eta\beta^2\sum_{i=r+1}^{A}\left(\sigma'_{\min}\delta(a_r, a_i; \theta^{(t)}) - \frac{1}{4}\delta(a_l, a_i; \theta^{(t)})\right) - \delta(a_r, a_l; \theta^{(t)})$$

$$= 4\eta\beta^2\left[\frac{1}{4}(l - 1)\delta(a_r, a_l; \theta^{(t)}) + \left(\frac{1}{4} - \sigma'_{\min}\right)\sum_{i=1}^{l-1}\delta(a_l, a_i; \theta^{(t)})\right] + 4\eta\beta^2\cdot\frac{1}{4}(r - l + 1)\delta(a_r, a_l; \theta^{(t)})$$

$$+ 4\eta\beta^2\left[\frac{1}{4}(A - r)\delta(a_r, a_l; \theta^{(t)}) + \left(\frac{1}{4} - \sigma'_{\min}\right)\sum_{i=r+1}^{A}\delta(a_i, a_r; \theta^{(t)})\right] - \delta(a_r, a_l; \theta^{(t)})$$

$$= \left(\eta\beta^2 A - 1\right)\delta(a_r, a_l; \theta^{(t)}) + 4\eta\beta^2\left(\frac{1}{4} - \sigma'_{\min}\right)\left(\sum_{i=1}^{l-1}\delta(a_l, a_i; \theta^{(t)}) + \sum_{i=r+1}^{A}\delta(a_i, a_r; \theta^{(t)})\right).$$

Now taking $\eta = \frac{1}{\beta^2 A}$, then we have

$$\delta(a_r, a_l; \theta^{(t+1)}) \leqslant (2 - 8\sigma'_{\min})\max_{a,a'}\delta(a, a'; \theta^{(t)}),$$

$$-\delta(a_r, a_l; \theta^{(t+1)}) \leqslant (1 - 4\sigma'_{\min})\max_{a,a'}\delta(a, a'; \theta^{(t)}).$$

Define

$$\gamma := 2 - 8\sigma'_{\min}$$

as the contraction factor, then

$$\left|\delta(a_r, a_l; \theta^{(t+1)})\right| \leqslant \gamma\max_{a,a'}\left|\delta(a, a'; \theta^{(t)})\right|. \tag{12}$$

Recall that we initialize $\theta^{(0)} = \vec{0}$. Next we use induction to verify that throughout the process ($t \geqslant 0$),

$$\left|\delta(a_r, a_l; \theta^{(t+1)})\right| \leqslant 0.214\gamma^t, \quad \text{and} \quad \left|\beta(\theta_a - \theta_{a'})^{(t+1)}\right| < 1.214. \tag{13}$$

For time $t = 0$, we have special versions: $r(a_1) \leqslant r(a_2) \leqslant \cdots \leqslant r(a_A)$.

$$\delta(a_r, a_l; \theta^{(1)}) = r(a_r) - r(a_l) - 4\eta\beta^2\sum_{i=1}^{A}(\Delta(a_r, a_i; \theta^{(0)}) - \Delta(a_l, a_i; \theta^{(0)}))$$

$$\overset{(i)}{=} r(a_r) - r(a_l) - 4\eta\beta^2\sum_{i=1}^{A}(\sigma(r(a_r) - r(a_i)) - \sigma(r(a_l) - r(a_i)))$$

$$\overset{(ii)}{\leqslant} r(a_r) - r(a_l) - 4\eta\beta^2\sum_{i=1}^{A}\sigma'(1)[r(a_r) - r(a_i) - (r(a_l) - r(a_i))]$$

$$= \left(1 - 4\eta\beta^2 A\sigma'(1)\right)(r(a_r) - r(a_l))$$

$$\overset{\text{(iii)}}{\leqslant} 0.214 \ ;$$

$$-\delta(a_r, a_l; \theta^{(1)}) = 4\eta\beta^2 \sum_{i=1}^{A} (\Delta(a_r, a_i; \theta^{(0)}) - \Delta(a_l, a_i; \theta^{(0)})) - (r(a_r) - r(a_l))$$

$$= 4\eta\beta^2 \sum_{i=1}^{A} (\sigma(r(a_r) - r(a_i)) - \sigma(r(a_l) - r(a_i))) - (r(a_r) - r(a_l))$$

$$\leqslant 4\eta\beta^2 \sum_{i=1}^{A} \frac{1}{4}[r(a_r) - r(a_i) - (r(a_l) - r(a_i))] - (r(a_r) - r(a_l))$$

$$= (\eta\beta^2 A - 1)(r(a_r) - r(a_l))$$

$$= 0 \ ,$$

where (i) is by $\theta^{(0)} = \vec{0}$; (ii) is by Equation (9) and $r(a_r) - r(a_i) \geqslant r(a_l) - r(a_i)$; (iii) is by $r(a_r) - r(a_l) \leqslant 1$. So $\left|\delta(a_r, a_l; \theta^{(1)})\right| \leqslant 0.214$, and $\left|\beta(\theta_a - \theta_{a'})_1\right| \leqslant \left|r(a) - r(a')\right| + \left|\delta(a_r, a_l; \theta^{(1)})\right| \leqslant 1.214$. Suppose for time $t - 1$, Equation (13) holds, then Equation (12) holds. So for time $t$,

$$\left|\delta(a_r, a_l; \theta^{(t+1)})\right| \leqslant \gamma \max_{a,a'} \left|\delta(a_r, a_l; \theta^{(t)})\right| \leqslant 0.214\gamma^t \leqslant 0.214 \ ,$$

and

$$\left|\beta(\theta_a - \theta_{a'})^{(t+1)}\right| \leqslant \left|r(a) - r(a')\right| + \left|\delta(a_r, a_l; \theta^{(t+1)})\right| \leqslant 1.214 \ .$$

Thus we have

$$\gamma = 2 - 8\sigma'_{\min} \leqslant 2 - 8\sigma'(1.214) < 0.588 \ ,$$

$$\left|\delta(a_r, a_l; \theta^{(T)})\right| \leqslant 0.588^T \ .$$

### A.1.2 CONSTRUCTION OF LOWER BOUND

Consider a three-armed bandit setting with rewards $r(a_1) = 0, r(a_2) = 1/3, r(a_3) = 1$ and any regularization coefficient $\beta \in \mathbb{R}_+$. The update rule satisfies:

$$\delta(a_2, a_1; \theta^{(t+1)}) = \delta(a_2, a_1; \theta^{(t)}) - 4\eta\beta^2 \left(2\Delta(a_2, a_1; \theta^{(t)}) + \Delta(a_3, a_1; \theta^{(t)}) - \Delta(a_3, a_2; \theta^{(t)})\right) \ , \tag{14}$$

$$\delta(a_3, a_2; \theta^{(t+1)}) = \delta(a_3, a_2; \theta^{(t)}) - 4\eta\beta^2 \left(2\Delta(a_3, a_2; \theta^{(t)}) + \Delta(a_3, a_1; \theta^{(t)}) - \Delta(a_2, a_1; \theta^{(t)})\right) \ , \tag{15}$$

$$\delta(a_3, a_1; \theta^{(t+1)}) = \delta(a_3, a_1; \theta^{(t)}) - 4\eta\beta^2 \left(2\Delta(a_3, a_1; \theta^{(t)}) + \Delta(a_3, a_2; \theta^{(t)}) + \Delta(a_2, a_1; \theta^{(t)})\right) \ .$$

Define $x_t := \delta(a_2, a_1; \theta^{(t)})$, and $y_t := \delta(a_3, a_2; \theta^{(t)})$. Clearly we have $\delta(a_3, a_1; \theta^{(t)}) = x_t + y_t$. We can perform Taylor expansion on Equations (14) and (15) and get

$$\begin{pmatrix} x_{t+1} \\ y_{t+1} \end{pmatrix} = \underbrace{\begin{pmatrix} 1 - 4\eta\beta^2(2\sigma'(1/3) + \sigma'(1)) & 4\eta(\sigma'(2/3) - \sigma'(1)) \\ 4\eta\beta^2(\sigma'(1/3) - \sigma'(1)) & 1 - 4\eta\beta^2(2\sigma'(2/3) + \sigma'(1)) \end{pmatrix}}_{:=B} \begin{pmatrix} x_t \\ y_t \end{pmatrix} + \eta\beta^2 \begin{pmatrix} u_t \\ v_t \end{pmatrix} \ , \tag{16}$$

where

$$|u_t| \leqslant \frac{4x_t^2 + 3y_t^2}{3\sqrt{3}} \leqslant x_t^2 + y_t^2 \ , \quad |v_t| \leqslant \frac{3x_t^2 + 4y_t^2}{3\sqrt{3}} \leqslant x_t^2 + y_t^2 \ . \tag{17}$$

Now we analyze the eigenvalues of $B$ under three scenarios.

1. If

$$0 < \eta\beta^2 < \frac{1}{4(2\sigma'(1/3) + \sigma'(1))} \approx 0.366 \ ,$$

then we have

$$\det(\lambda I - B) = \lambda^2 - (B_{11} + B_{22})\lambda + \underbrace{B_{11}B_{22}}_{\leqslant (B_{11}+B_{22})^2/4} - \underbrace{B_{12}B_{21}}_{>0} \ .$$

2. If

$$\frac{1}{4(2\sigma'(1/3) + \sigma'(1))} \leqslant \eta\beta^2 \leqslant \frac{1}{4(2\sigma'(2/3) + \sigma'(1))} \approx 0.388 \ ,$$

then we have

$$\det(\lambda I - B) = \lambda^2 - (B_{11} + B_{22})\lambda + \underbrace{B_{11}B_{22}}_{\leqslant 0} - \underbrace{B_{12}B_{21}}_{>0} \ .$$

3. If

$$\frac{1}{4(2\sigma'(2/3) + \sigma'(1))} < \eta\beta^2 < \frac{1}{2(2\sigma'(1/3) + \sigma'(2/3))} \approx 0.704 \ ,$$

then we have

$$\det(\lambda I - B) = \lambda^2 - (B_{11} + B_{22})\lambda + \underbrace{B_{11}B_{22}}_{\leqslant (B_{11}+B_{22})^2/4} - \underbrace{B_{12}B_{21}}_{>0} \ .$$

Therefore $B$ has two different eigenvalues $\lambda_1, \lambda_2 \in (-1,0) \cup (0,1)$, with normalized eigenvectors $w_1, w_2$. Clearly $w_{ij} \in (-1,0) \cup (0,1)$, $\forall i,j \in \{1,2\}$. Then we define $\lambda_{\max} := \max(|\lambda_1|, |\lambda_2|)$, $\lambda_{\min} := \min(|\lambda_1|, |\lambda_2|)$, Now perform basis transformation with new basis $(w_1, w_2)$. Thus Equation (16) can be rewritten as

$$\begin{pmatrix} p_{t+1} \\ q_{t+1} \end{pmatrix} = \begin{pmatrix} \lambda_1 & 0 \\ 0 & \lambda_2 \end{pmatrix} \begin{pmatrix} p_t \\ q_t \end{pmatrix} + \begin{pmatrix} u'_t \\ v'_t \end{pmatrix} \ ,$$

Let $w'_1, w'_2$ be the inverse basis, and define $\alpha := \max\limits_{i,j\in\{1,2\}} |w'_{ij}|$, and $\epsilon := \min\left(\lambda_{\min}, 1 - \lambda_{\max}\right)/(64\alpha^2)$. Now initialize $|x_0|, |y_0| \in (0,\epsilon)$. Then we have $\max\limits_{i,j\in\{1,2,3\}} |\delta(a_i, a_j; \theta^{(0)})| \leqslant 2\epsilon$. Therefore

$$|p_0|, |q_0| \overset{(i)}{\leqslant} 2\alpha\epsilon \ ,$$

and

$$|u'_t|, |v'_t| \overset{(ii)}{\leqslant} 2\alpha(x_t^2 + y_t^2)$$
$$\overset{(iii)}{\leqslant} 4\alpha(p_t^2 + q_t^2) \ .$$

(i) and (ii) comes from the fact that $p_t = w'_{11}x_t + w'_{21}y_t$ and $q_t = w'_{12}x_t + w'_{22}y_t$, and Equation (17); (iii) is from the fact that $x_t = w_{11}p_t + w_{21}q_t$ and $y_t = w_{12}p_t + w_{22}q_t$, and Cauchy-Schwarz inequality. Now we have

$$|p_{t+1}| + |q_{t+1}| \leqslant [\lambda_{\max} + 8\alpha(|p_t| + |q_t|)](|p_t| + |q_t|)$$
$$\overset{(iv)}{\leqslant} \left(\lambda_{\max} + 32\alpha^2\epsilon\right)(|p_t| + |q_t|)$$
$$\leqslant \frac{1 + \lambda_{\max}}{2}(|p_t| + |q_t|) \ .$$
$$|p_{t+1}| + |q_{t+1}| \geqslant [\lambda_{\min} - 8\alpha(|p_t| + |q_t|)](|p_t| + |q_t|)$$
$$\overset{(v)}{\geqslant} \left(\lambda_{\min} - 32\alpha^2\epsilon\right)(|p_t| + |q_t|)$$
$$\geqslant \frac{\lambda_{\min}}{2}(|p_t| + |q_t|) \ ,$$

where (iv) and (v) are based on simple induction that $|p_t| + |q_t|$ will not increase. And it thus indicates that $\max(|x_t|, |y_t|)$ can at most achieve linear convergence when $\eta\beta^2 \leqslant \frac{2}{A} \approx 0.667$.

### A.2 THEOREM 3: QUADRATIC CONVERGENCE OF EXACT `DPO-Mix-R`

We study DPO with a mixture of fixed samplers: $Z^+ Z^- \cdot \pi^{\mathsf{s1}} \times \pi^{\mathsf{s2}} + A^2 \cdot \mathsf{Uniform}(\mathcal{A}) \times \mathsf{Uniform}(\mathcal{A})$, where $Z^+ = \sum_a \exp(r(a))$, $\pi^{\mathsf{s1}}(a) = \exp(r(a))/Z^+$ and $Z^- = \sum_a \exp(-r(a))$, $\pi^{\mathsf{s2}}(A) = \exp(-r(a))/Z^-$. We have

$$\alpha_1 \mathcal{L}_1(\theta) + \alpha_2 \mathcal{L}_2(\theta)$$

$$= -\sum_{a,a'} \left( A^2 \cdot \frac{1}{A^2} + Z^+ Z^- \cdot \pi^{\mathsf{s1}}(a)\pi^{\mathsf{s2}}(a') \right) \left[ p^\star(a > a') \log \sigma \left( \beta \log \frac{\pi_\theta(a)}{\pi_\theta(a')} \right) + p^\star(a' > a) \log \sigma \left( \beta \log \frac{\pi_\theta(a')}{\pi_\theta(a)} \right) \right]$$

$$= -\sum_{a,a'} (\exp(r(a) - r(a')) + 1) \left[ p^\star(a > a') \log \sigma \left( \beta \log \frac{\pi_\theta(a)}{\pi_\theta(a')} \right) + p^\star(a' > a) \log \sigma \left( \beta \log \frac{\pi_\theta(a')}{\pi_\theta(a)} \right) \right],$$

$$\alpha_1 \nabla_\theta \mathcal{L}_1(\theta) + \alpha_2 \nabla_\theta \mathcal{L}_2(\theta)$$

$$= -\beta \sum_{a,a'} (\exp(r(a) - r(a')) + 1)\Delta(a, a'; \theta)(\mathbb{1}_a - \mathbb{1}_{a'})$$

$$= -\beta \sum_{a,a'} (\exp(r(a) - r(a')) + \exp(r(a') - r(a)) + 2)\Delta(a, a'; \theta)\mathbb{1}_a$$

$$= -\beta \sum_{a,a'} \frac{\Delta(a, a'; \theta)}{\sigma'(r(a) - r(a'))} \mathbb{1}_a. \tag{18}$$

Equation (18) reduces to

$$\alpha_1 \nabla_{\theta_a} \mathcal{L}_1(\theta) + \alpha_2 \nabla_{\theta_a} \mathcal{L}_2(\theta) = -\beta \sum_{a'} \frac{\Delta(a, a'; \theta)}{\sigma'(r(a) - r(a'))}.$$

Fix parameter $\theta$. For any action pair $a, a'$, through Taylor expansion we have that

$$\Delta(a, a'; \theta) = \sigma'(r(a) - r(a'))\delta(a, a'; \theta) - \frac{\sigma''(\xi_{\mathsf{R}}(a, a'; \theta))}{2}\delta(a, a'; \theta)^2,$$

where $\xi_{\mathsf{R}}(a, a'; \theta)$ is between $r(a) - r(a')$ and $\beta(\theta_a - \theta_{a'})$. We have that at time step $t$, for any action pair $(a, a')$,

$$\delta(a, a'; \theta^{(t+1)}) = \delta(a, a'; \theta^{(t)}) - \eta\beta^2 \sum_{a''} \left( \frac{\Delta(a, a''; \theta^{(t)})}{\sigma'(r(a) - r(a''))} - \frac{\Delta(a', a''; \theta^{(t)})}{\sigma'(r(a') - r(a''))} \right)$$

$$= \delta(a, a'; \theta^{(t)}) - \eta\beta^2 \sum_{a''} (\delta(a, a''; \theta^{(t)}) - \delta(a', a''; \theta^{(t)}))$$

$$+ \frac{\eta\beta^2}{2} \sum_{a''} \left( \frac{\sigma''(\xi_{\mathsf{R}}(a, a''; \theta^{(t)}))}{\sigma'(r(a) - r(a''))}\delta(a, a''; \theta^{(t)})^2 - \frac{\sigma''(\xi_{\mathsf{R}}(a', a''; \theta^{(t)}))}{\sigma'(r(a') - r(a''))}\delta(a', a''; \theta^{(t)})^2 \right)$$

$$= (1 - \eta\beta^2 A)\delta(a, a'; \theta^{(t)})$$

$$+ \frac{\eta\beta^2}{2} \sum_{a''} \left( \frac{\sigma''(\xi_{\mathsf{R}}(a, a''; \theta^{(t)}))}{\sigma'(r(a) - r(a''))}\delta(a, a''; \theta^{(t)})^2 - \frac{\sigma''(\xi_{\mathsf{R}}(a', a''; \theta^{(t)}))}{\sigma'(r(a') - r(a''))}\delta(a', a''; \theta^{(t)})^2 \right).$$

From the range of $r$, we know that $\sigma'(r(a) - r(a')) \geqslant \sigma'(1) > 0.196$. We have $|\sigma''(\xi_{\mathsf{R}}(a, a''; \theta^{(t)}))| \leqslant \sigma''_{\max} := \sup_{0 \leqslant x \leqslant 1} x(1-x)(1-2x) = 1/(6\sqrt{3}) < 0.097$. Set

$$\eta = \frac{1}{\beta^2 A},$$

then

$$\left| \delta(a, a'; \theta^{(t+1)}) \right| \leqslant \frac{1}{2A} \sum_{a''} \left( \frac{\sigma''_{\max}}{\sigma'(1)}\delta(a, a''; \theta^{(t)})^2 + \frac{\sigma''_{\max}}{\sigma'(1)}\delta(a', a''; \theta^{(t)})^2 \right)$$

$$\leqslant \frac{\sigma''_{\max}}{\sigma'(1)} \max_{a,a'} \delta(a, a'; \theta^{(t)})^2$$

$$< \frac{1}{2} \max_{a,a'} \delta(a, a'; \theta^{(t)})^2 \ .$$

Since $\max_{a,a'} \left| \delta(a, a'; \theta^{(0)}) \right| \leqslant 1$, we can show a quadratic convergence for this regime:

$$\left| \delta(a, a'; \theta^{(t)}) \right| \leqslant 0.5^{2^t - 1} \ .$$

## A.3 THEOREM 4: QUADRATIC CONVERGENCE OF EXACT `DPO-Mix-P`

We study DPO with a mixture of online samplers (with gradient stopped) and uniform samplers:
$Z^+ Z^- \cdot \pi^{\mathsf{s}1} \times \pi^{\mathsf{s}2} + A^2 \cdot \mathsf{Uniform}(\mathcal{A}) \times \mathsf{Uniform}(\mathcal{A})$ , where $Z^+ = \sum_a \exp(\beta\theta_a)$ , $\pi^{\mathsf{s}1}(a) = \exp(\beta\theta_a)/Z^+$ and $Z^- = \sum_a \exp(-\beta\theta_a)$ , $\pi^{\mathsf{s}2}(a) = \exp(-\beta\theta_a)/Z^-$. Samely we have

$$\alpha_1 \nabla_{\theta_a} \mathcal{L}_1(\theta) + \alpha_2 \nabla_{\theta_a} \mathcal{L}_2(\theta) = -\beta \sum_{a'} \frac{\Delta(a, a'; \theta)}{\sigma'(\beta(\theta_a - \theta_{a'}))} \ .$$

Fix parameter $\theta$. For any action pair $a, a'$, through Taylor expansion we have that

$$\Delta(a, a'; \theta) = \sigma'(\beta(\theta_a - \theta_{a'}))\delta(a, a'; \theta) + \frac{\sigma''(\xi_{\mathsf{P}}(a, a'; \theta))}{2}\delta(a, a'; \theta)^2 \ ,$$

where $\xi_{\mathsf{P}}(a, a'; \theta)$ is between $r(a) - r(a')$ and $\beta(\theta_a - \theta_{a'})$. We have that at time step $t$, for any action pair $(a, a')$,

$$
\begin{aligned}
\delta(a, a'; \theta^{(t+1)}) &= \delta(a, a'; \theta^{(t)}) - \eta\beta^2 \sum_{a''} \left( \frac{\Delta(a, a''; \theta^{(t)})}{\sigma'(\beta(\theta_a - \theta_{a''})^{(t)})} - \frac{\Delta(a', a''; \theta^{(t)})}{\sigma'(\beta(\theta_{a'} - \theta_{a''})^{(t)})} \right) \\
&= \delta(a, a'; \theta^{(t)}) - \eta\beta^2 \sum_{a''} (\delta(a, a''; \theta^{(t)}) - \delta(a', a''; \theta^{(t)})) \\
&\quad - \frac{\eta\beta^2}{2} \sum_{a''} \left( \frac{\sigma''(\xi_{\mathsf{P}}(a, a''; \theta^{(t)}))}{\sigma'(\beta(\theta_a - \theta_{a''})^{(t)})} \delta(a, a''; \theta^{(t)})^2 - \frac{\sigma''(\xi_{\mathsf{P}}(a', a''; \theta^{(t)}))}{\sigma'(\beta(\theta_{a'} - \theta_{a''})^{(t)})} \delta(a', a''; \theta^{(t)})^2 \right) \\
&= (1 - \eta\beta^2 A)\delta(a, a'; \theta^{(t)}) \\
&\quad - \frac{\eta\beta^2}{2} \sum_{a''} \left( \frac{\sigma''(\xi_{\mathsf{P}}(a, a''; \theta^{(t)}))}{\sigma'(\beta(\theta_a - \theta_{a''})^{(t)})} \delta(a, a''; \theta^{(t)})^2 - \frac{\sigma''(\xi_{\mathsf{P}}(a', a''; \theta^{(t)}))}{\sigma'(\beta(\theta_{a'} - \theta_{a''})^{(t)})} \delta(a', a''; \theta^{(t)})^2 \right) \ .
\end{aligned}
$$

We still first claim that $\sigma'(\beta(\theta_a - \theta_{a'})_t) \geqslant \sigma'_{\min}$, and will bound it later. We have $\left| \sigma''(\xi_{\mathsf{P}}(a, a''; \theta^{(t)})) \right| \leqslant \sigma''_{\max} < 0.097$. Set

$$\eta = \frac{1}{\beta^2 A} \ ,$$

then

$$\left| \delta(a, a'; \theta^{(t+1)}) \right| \leqslant \frac{\sigma''_{\max}}{2A\sigma'_{\min}} \sum_{a''} (\delta(a, a''; \theta^{(t)})^2 + \delta(a', a''; \theta^{(t)})^2)$$

$$\leqslant \frac{\sigma''_{\max}}{\sigma'_{\min}} \max_{a,a'} \delta(a, a'; \theta^{(t)})^2 \ .$$

At time step $t = 0$ we have $\sigma'(\beta(\theta_a - \theta_{a'})^{(0)}) = \sigma'(0) = 0.25$ and $\max_{a,a'} \left| \delta(a, a'; \theta^{(0)}) \right| \leqslant 1$ , so

$$\max_{a,a'} \left| \delta(a, a'; \theta^{(1)}) \right| < 0.388 \ .$$

By simple induction, we have that

$$\sigma'_{\min} \geqslant \sigma'(1 + \max_{a,a'} \left| \delta(a, a'; \theta^{(t)}) \right|) \geqslant \sigma'(1.388) > 0.159 \ ,$$

$$\max_{a,a'} \left| \delta(a, a'; \theta^{(t+1)}) \right| \leqslant \frac{0.097}{0.159} \max_{a,a'} \delta(a, a'; \theta^{(t)})^2 < 0.611 \max_{a,a'} \delta(a, a'; \theta^{(t)})^2 \ .$$

which is a quadratic convergence:

$$\left| \delta(a, a'; \theta^{(t)}) \right| \leqslant 0.611^{2^t - 1} \ .$$

# B    PROOF OF CONVERGENCE RATES OF EMPIRICAL DPO

For notational ease, we make the following definitions throughout this section:

$$\Delta(a, a'; \theta) := \sigma(r(a) - r(a')) - \sigma(\beta(\theta_a - \theta_{a'})) ,$$
$$\delta(a, a'; \theta) := r(a) - r(a') - \beta(\theta_a - \theta_{a'}) .$$

This section conforms to Definition 2. Denote the filtration $\mathcal{F}_t$ as all the samples on and before time step $t$.

## B.1    TECHNICAL LEMMA

**Lemma 2** (Lemma 1.4 in Philippe Rigollet (2015)). *Let $X$ be a random variable such that*

$$\mathbb{P}[|X| > t] \leqslant 2\exp\left(-\frac{t^2}{2\sigma^2}\right) ,$$

*then for any positive integer $k \geqslant 2$,*

$$\mathbb{E}[|X|^k] \leqslant (\sigma e^{1/e}\sqrt{k})^k ,$$

*and*

$$\mathbb{E}[|X|] \leqslant \sigma\sqrt{2\pi} .$$

## B.2    THEOREM 5: CONVERGENCE OF EMPIRICAL `DPO-Mix-R`

Similar to Appendix A.2, at time step $t$, conditioned on $\mathcal{F}_t$, we have that for any action pair $(a, a')$,

$$\mathbb{E}[(G_a - G_{a'})^{(t)}] = -\beta A\delta(a, a'; \theta^{(t)})$$
$$- \frac{\beta}{2}\underbrace{\sum_{a''}\left(\frac{\sigma''(\xi_R(a, a''; \theta^{(t)}))}{\sigma'(r(a) - r(a''))}\delta(a, a''; \theta^{(t)})^2 - \frac{\sigma''(\xi_R(a', a''; \theta^{(t)}))}{\sigma'(r(a') - r(a''))}\delta(a', a''; \theta^{(t)})^2\right)}_{=:N_t(a,a')},$$

$$|N_t(a, a')| < \frac{1}{2}\sum_{a''}(\delta(a, a''; \theta^{(t)})^2 + \delta(a', a''; \theta^{(t)})^2) . \tag{19}$$

From Definition 2 and Lemma 2, we have that

$$\mathbb{E}\left[\left|\frac{G_a^{(t)} - \mathbb{E}[G_a^{(t)}]}{\beta A}\right|^k\right] \leqslant (3\sigma\sqrt{k})^k .$$

Therefore, from Minkowski inequality,

$$\mathbb{E}\left[\left|\frac{(G_a - G_{a'})^{(t)} - \mathbb{E}[(G_a - G_{a'})^{(t)}]}{\beta A}\right|^k\right] \leqslant (6\sigma\sqrt{k})^k .$$

Now we take $\eta = 1/(\beta^2 A)$, then by taking expectation conditioning on $\mathcal{F}_t$ we obtain

$$\mathbb{E}[\delta(a, a'; \theta^{(t+1)})^{2n}] = \mathbb{E}[(\delta(a, a'; \theta^{(t)}) + \eta\beta(G_a - G_{a'}))^{2n}]$$
$$= \mathbb{E}[[\delta(a, a'; \theta^{(t)}) + \eta\beta\mathbb{E}[G_a - G_{a'}] + \eta\beta(G_a - G_{a'} - \mathbb{E}[G_a - G_{a'}])]^{2n}]$$
$$= \sum_{k=0}^{2n}\binom{2n}{k}(\delta(a, a'; \theta^{(t)}) + \eta\beta\mathbb{E}[G_a - G_{a'}])^{2n-k} \cdot (\eta\beta)^k\mathbb{E}[(G_a - G_{a'} - \mathbb{E}[G_a - G_{a'}])^k]$$
$$\overset{(i)}{=} \sum_{k=0}^{2n}\binom{2n}{k}\left(-\frac{1}{2A}N_t(a, a')\right)^{2n-k} \cdot \frac{1}{(\beta A)^k}\mathbb{E}[(G_a - G_{a'} - \mathbb{E}[G_a - G_{a'}])^k]$$
$$\leqslant \sum_{k=0}^{2n}\binom{2n}{k}\left(\frac{1}{2A}|N_t(a, a')|\right)^{2n-k}(6\sigma\sqrt{k})^k ,$$

where (i) is by substituting $\eta = 1/(\beta^2 A)$.

Further taking expectation over $\mathcal{F}_t$, we have

$$
\mathbb{E}[\delta(a, a'; \theta^{(t+1)})^{2n}]
$$

$$
\leq \sum_{k=0}^{2n} \frac{\binom{2n}{k}}{(2A)^{2n-k}} (6\sigma\sqrt{k})^k \mathbb{E}[|N_t|^{2n-k}(a, a')]
$$

$$
\overset{(i)}{\leq} \sum_{k=0}^{2n} \frac{\binom{2n}{k}}{(2A)^{2n-k}} (6\sigma\sqrt{k})^k \cdot \frac{1}{2^{2n-k}} \mathbb{E}\left[\left[\sum_{a''}(\delta(a, a''; \theta^{(t)})^2 + \delta(a', a''; \theta^{(t)})^2)\right]^{2n-k}\right]
$$

$$
\overset{(ii)}{\leq} \sum_{k=0}^{2n} \frac{\binom{2n}{k}}{(2A)^{2n-k}} (6\sigma\sqrt{k})^k \cdot \frac{1}{2^{2n-k}} \cdot (2A)^{2n-k-1} \sum_{a''}(\mathbb{E}[\delta(a, a''; \theta^{(t)})^{4n-2k}] + \mathbb{E}[\delta(a', a''; \theta^{(t)})^{4n-2k}])
$$

$$
\leq \sum_{k=0}^{2n} \binom{2n}{k} (6\sigma\sqrt{k})^k \cdot \frac{1}{2^{2n-k}} \max_{a_1, a_2} \mathbb{E}[\delta(a_1, a_2; \theta^{(t)})^{4n-2k}]
$$

$$
\leq \sum_{k=0}^{2n} \binom{2n}{k} (6\sigma\sqrt{n})^k \cdot \frac{1}{2^{2n-k}} \max_{a_1, a_2} \mathbb{E}[\delta(a_1, a_2; \theta^{(t)})^{4n-2k}] \, ,
$$

where (i) is by Equation (19); (ii) is by Hölder inequality.

Take $T = \lfloor \log(1/\sigma) \rfloor$. When $\sigma \leq 1/576 < 0.00174$, we will show that $\forall n, t \in \mathbb{N}$ such that $n \cdot 2^t \leq 1/\sigma$,

$$
\mathbb{E}[\delta(a, a'; \theta^{(t)})^{2n}] \leq \left(12\sqrt{n}\sigma + \frac{1}{2^t}\right)^{2n} .
$$

This can be proved using induction on $t$. For $t \leq 1$, we have that for any $n$:

$$
\mathbb{E}[\delta(a, a'; \theta^{(0)})^{2n}] \leq 1 \, ,
$$

$$
\mathbb{E}[\delta(a, a'; \theta^{(1)})^{2n}] \leq \left(6\sqrt{n}\sigma + \frac{1}{2}\right)^{2n} .
$$

For $t = 2$ and $n \leq 1/(4\sigma)$,

$$
\mathbb{E}[\delta(a, a'; \theta^{(2)})^{2n}] \leq \sum_{k=0}^{2n} \binom{2n}{k} (6\sigma\sqrt{n})^k \cdot \frac{1}{2^{2n-k}} \left(6\sqrt{2n}\sigma + \frac{1}{2}\right)^{4n-2k}
$$

$$
\leq \left(6\sqrt{n}\sigma + \frac{(6\sqrt{2n}\sigma + \frac{1}{2})^2}{2}\right)^{2n}
$$

$$
= \left(36n\sigma^2 + (6 + 3\sqrt{2})\sqrt{n}\sigma + \frac{1}{8}\right)^{2n}
$$

$$
\overset{(i)}{\leq} \left(12\sqrt{n}\sigma + \frac{1}{2^2}\right)^{2n} ,
$$

where (i) is by plugging in the range of $n$ and $\sigma$. Suppose the arguments holds for $t \geq 2$, then

$$
\mathbb{E}[\delta(a, a'; \theta^{(t+1)})^{2n}] \leq \sum_{k=0}^{2n} \binom{2n}{k} (6\sigma\sqrt{n})^k \cdot \frac{1}{2^{2n-k}} \left(12\sqrt{2n}\sigma + \frac{1}{2^t}\right)^{4n-2k}
$$

$$
= \left[6\sqrt{n}\sigma + \frac{\left(12\sqrt{2n}\sigma + \frac{1}{2^t}\right)^2}{2}\right]^{2n}
$$

$$
\leq \left[\left(6 + \frac{12\sqrt{2}}{2^t}\right)\sqrt{n}\sigma + 144n\sigma^2 + \frac{1}{2^{2t+1}}\right]^{2n}
$$

$$\overset{(i)}{\leqslant} \left[ \left(6 + \frac{12\sqrt{2}}{2^t}\right) \sqrt{n}\sigma + \frac{288\sigma + \frac{1}{2^t}}{2^{t+1}} \right]^{2n}$$

$$\overset{(ii)}{\leqslant} \left(12\sqrt{n}\sigma + \frac{1}{2^{t+1}}\right)^{2n} ,$$

where (i) is by $n \leqslant 1/(\sigma \cdot 2^t)$; (ii) is by $t \geqslant 2$ and the range of $\sigma$.

Therefore, we have for $\sigma \leqslant 1/576$ and $T = \lfloor \log(1/\sigma) \rfloor > \log(1/\sigma) - 1$,

$$\sqrt{\mathbb{E}[\delta(a, a'; \theta^{(T)})^2]} \leqslant 12\sigma + \frac{1}{2^T} < 14\sigma .$$

## B.3 THEOREM 6: CONVERGENCE OF EMPIRICAL DPO-Mix-P*

Here we use the joint probability weights $\psi(a, a') \propto \exp(z(a, a'))$ such that $z(a, a') = -z(a', a)$ and let $Z := \sum_{a,a'} \exp(z(a, a'))$:

$$\alpha_1 \mathcal{L}_1(\theta) + \alpha_2 \mathcal{L}_2(\theta)$$
$$= -\sum_{a,a'} \mathsf{sg}\left(A^2 \cdot \frac{1}{A^2} + Z \cdot \psi(a, a')\right) \left[p^\star(a > a') \log \sigma \left(\beta \log \frac{\pi_\theta(a)}{\pi_\theta(a')}\right) + p^\star(a' > a) \log \sigma \left(\beta \log \frac{\pi_\theta(a')}{\pi_\theta(a)}\right)\right] ,$$

$$\alpha_1 \nabla_\theta \mathcal{L}_1(\theta) + \alpha_2 \nabla_\theta \mathcal{L}_2(\theta)$$
$$= -\beta \sum_{a,a'} \left(\exp(z(a, a')) + 1\right) \Delta(a, a'; \theta)(\mathbb{1}_a - \mathbb{1}_{a'})$$
$$= -\beta \sum_{a,a'} \left(\exp(z(a, a')) + \exp(-z(a, a')) + 2\right) \Delta(a, a'; \theta)\mathbb{1}_a$$
$$= -\beta \sum_{a,a'} \frac{\Delta(a, a'; \theta)}{\sigma'(z(a, a'))} \mathbb{1}_a . \tag{20}$$

Equation (20) reduces to

$$\nabla_{\theta_a} \mathcal{L}(\theta) = -\beta \sum_{a'} \frac{\Delta(a, a'; \theta)}{\sigma'(z(a, a'))} .$$

Fix parameter $\theta$. For any action pair $a, a'$, through Taylor expansion we have that

$$\Delta(a, a'; \theta) = (\sigma(r(a) - r(a')) - \sigma(z(a, a'))) - (\sigma(\beta(\theta_a - \theta_{a'})) - \sigma(z(a, a'))))$$
$$= [\sigma'(z(a, a'))(r(a) - r(a') - z(a, a')) + \frac{\sigma''(\xi_1(a, a'; \theta))}{2}(r(a) - r(a') - z(a, a'))^2]$$
$$- \{\sigma'(z(a, a'))[\beta(\theta_a - \theta_{a'}) - z(a, a')] + \frac{\sigma''(\xi_2(a, a'; \theta))}{2}[\beta(\theta_a - \theta_{a'}) - z(a, a')]^2\}$$
$$= \sigma'(z(a, a'))\delta(a, a'; \theta) + \frac{\sigma''(\xi_1(a, a'; \theta))}{2}(r(a) - r(a') - z(a, a'))^2$$
$$- \frac{\sigma''(\xi_2(a, a'; \theta))}{2}[\beta(\theta_a - \theta_{a'}) - z(a, a')]^2 ,$$

where $\xi_1(a, a'; \theta)$ is between $r(a) - r(a')$ and $z(a, a')$, and $\xi_2(a, a'; \theta)$ is between $z(a, a')$ and $\beta(\theta_a - \theta_{a'})$ .

If we set

$$z(a, a') = \begin{cases} 1, & \text{if } \beta(\theta_a - \theta_{a'}) > 1 , \\ -1, & \text{if } \beta(\theta_a - \theta_{a'}) < -1 , \\ \beta(\theta_a - \theta_{a'}), & \text{otherwise} , \end{cases}$$

then we can conclude that

$$[r(a) - r(a') - z(a, a')]^2 + [\beta(\theta_a - \theta_{a'}) - z(a, a')]^2 \leqslant \delta(a, a'; \theta)^2 .$$

Note that this construction satisfies $z(a, a') = -z(a', a)$ . We have that at time step $t$, conditioning on $\mathcal{F}_t$, for any action pair $(a, a')$ ,

$$\mathbb{E}[\delta(a, a'; \theta^{(t+1)})]$$

$$= \delta(a, a'; \theta^{(t)}) - \eta\beta^2 \sum_{a''} \left( \frac{\Delta(a, a''; \theta^{(t)})}{\sigma'(z(a, a''))} - \frac{\Delta(a', a''; \theta^{(t)})}{\sigma'(z(a', a''))} \right)$$

$$= \delta(a, a'; \theta^{(t)}) - \eta\beta^2 \sum_{a''} (\delta(a, a''; \theta^{(t)}) - \delta(a', a''; \theta^{(t)}))$$

$$- \frac{\eta\beta^2}{2} \sum_{a''} \left\{ \frac{\sigma''(\xi_1(a, a''; \theta^{(t)}))}{\sigma'(z(a, a''))}(r(a) - r(a'') - z(a, a''))^2 - \frac{\sigma''(\xi_2(a, a''; \theta^{(t)}))}{\sigma'(z(a, a''))}[\beta(\theta_a - \theta_{a''})^{(t)} - z(a, a'')]^2 \right\}$$

$$+ \frac{\eta\beta^2}{2} \sum_{a''} \left\{ \frac{\sigma''(\xi_1(a', a''; \theta^{(t)}))}{\sigma'(z(a', a''))}(r(a') - r(a'') - z(a', a''))^2 - \frac{\sigma''(\xi_2(a', a''; \theta^{(t)}))}{\sigma'(z(a', a''))}[\beta(\theta_{a'} - \theta_{a''})^{(t)} - z(a', a'')]^2 \right\}$$

$$= (1 - \eta\beta^2 A)\delta(a, a'; \theta^{(t)})$$

$$- \frac{\eta\beta^2}{2} \sum_{a''} \left\{ \frac{\sigma''(\xi_1(a, a''; \theta^{(t)}))}{\sigma'(z(a, a''))}(r(a) - r(a'') - z(a, a''))^2 - \frac{\sigma''(\xi_2(a, a''; \theta^{(t)}))}{\sigma'(z(a, a''))}[\beta(\theta_a - \theta_{a''})^{(t)} - z(a, a'')]^2 \right\}$$

$$+ \frac{\eta\beta^2}{2} \sum_{a''} \left\{ \frac{\sigma''(\xi_1(a', a''; \theta^{(t)}))}{\sigma'(z(a', a''))}(r(a') - r(a'') - z(a', a''))^2 - \frac{\sigma''(\xi_2(a', a''; \theta^{(t)}))}{\sigma'(z(a', a''))}[\beta(\theta_{a'} - \theta_{a''})^{(t)} - z(a', a'')]^2 \right\} \ .$$

Set

$$\eta = \frac{1}{\beta^2 A} \ ,$$

then

$$\mathbb{E}\left|\delta(a, a'; \theta^{(t+1)})\right| \leqslant \frac{\sigma''_{\max}}{2A\sigma'(1)} \sum_{a''} \{(r(a) - r(a'') - z(a, a''))^2 + [\beta(\theta_a - \theta_{a''})^{(t)} - z(a, a'')]^2$$

$$+ (r(a') - r(a'') - z(a', a''))^2 + [\beta(\theta_{a'} - \theta_{a''})^{(t)} - z(a', a'')]^2\}$$

$$< \frac{1}{2A} \cdot \underbrace{\frac{1}{2} \sum_{a''} (\delta(a, a''; \theta^{(t)})^2 + \delta(a', a''; \theta^{(t)})^2)}_{=: \widetilde{N}_t(a, a')} \ .$$

Here $\sigma''_{\max} = 1/(6\sqrt{3}) < 0.097$ as before and $\sigma'(1) > 0.196$.

Follow the same steps as in Appendix B.2, we have that for $\sigma \leqslant 1/576$ and $T = \lfloor \log(1/\sigma) \rfloor$,

$$\sqrt{\mathbb{E}[\delta(a, a'; \theta^{(T)})^2]} < 14\sigma \ .$$

## C  IMPLEMENTATION DETAILS

**Codebases & Datasets.**  Our codebase is mainly based on the pipeline of Xiong et al. (2024); Dong et al. (2024) (`https://github.com/RLHFlow/Online-RLHF`), and has referred to Shi et al. (2024) (`https://github.com/srzer/MOD`) for the implementation of logit mixing. For Safe-RLHF, we adopt a 10k subset of Ji et al. (2023a) (`https://huggingface.co/datasets/PKU-Alignment/PKU-SafeRLHF`) for training, and a 2k subset as test set; For Iterative-Prompt, we adopt a 10k subset of Xiong et al. (2024); Dong et al. (2024) (`RLHFlow/iterative-prompt-v1-iter1-20K`) for training, and a 2k subset as test set.

**Policy models & Reward model.**  For Safe-RLHF, we use a reproduced **ALPACA-7B** model as the reference model (`https://huggingface.co/PKU-Alignment/alpaca-7b-reproduced`). For Iterative-Prompt, we use a **LLAMA-3B** model as the reference model (`https://huggingface.co/openlm-research/open_llama_3b_v2`). We use the reward model of Dong et al. (2023) (`https://huggingface.co/sfairXC/FsfairX-LLaMA3-RM-v0.1`) for two tasks.

**Implementation of mixed samplers and reward margin.**  Given the mixing ratio set as $1-\alpha : \alpha$, for each prompt, we add a generated pair from ① with probability $1-\alpha$, and from ② with probability $\alpha$. As we stated in Eq. (8), $\alpha$ can be approximated as $\frac{\exp(r)+\exp(-r)}{\exp(r)+\exp(-r)+2}$. As for the reward margin $r_{\max}$, unlike common practice as Xiong et al. (2024); Dong et al. (2024) setting $r_{\max} = +\infty$, we set $r_{\max} = 4$ for Safe-RLHF and $r_{\max} = 1$ for Iterative-Prompt, to better align with the assumed BT-model setting. Therefore, we use $\alpha = 0.7$ for the former and $\alpha = 1$ for the latter. We did not extensively tune these hyperparameters, as our focus has been on validation of theoretical claims.

**Hyperparameters.**  The hyperparameters are borrowed from Dong et al. (2024) with minimal modifications.  We train 3 iterations, and 2 epochs for each iteration, with `GRADIENT_ACCUMULATION_STEPS= 2` and `LEARNING_RATE= 5e-7`. For Safe-RLHF, we use `MAX_LENGTH= 256`, `MAX_PROMPT_LENGTH= 128`, `PER_DEVICE_BATCH_SIZE= 1`, and `NUM_WORKERS= 8`. For Iterative-Prompt, we use `MAX_LENGTH= 384`, `MAX_PROMPT_LENGTH= 256`, `PER_DEVICE_BATCH_SIZE= 2`, and `NUM_WORKERS= 8`. During generation for training, we set temperature $\tau = 0.7$, while during evaluation we set $\tau = 0.1$.

## D  SUPPLEMENTARY RESULTS

### D.1  MORE NUMERICAL SIMULATIONS

**Configurations.**  The numerical simulations are conducted on 20-arm bandits. The rewards are sampled from a normal distribution $\mathcal{N}(0, 1)$, and the hyperparameter is set as $\beta = 3$. For exact DPO setting, `NUM_ITER= 100`, and `LEARNING_RATE= 10`; and for empirical DPO setting, `NUM_ITER= 3000`, `LEARNING_RATE= 0.05`.

**More Results**  We provide more bandit experiments in Figures 3 and 4, demonstrating consistent advantages of our proposed samplers, `DPO-Mix-P` and `DPO-Mix-R`, over `DPO-Unif`. Besides, we conduct ablation experiments on the mixed components, ① and ②, in `DPO-Mix-P` and `DPO-Mix-R`, and results shown in Figures 5 and 6 indicate that the ② component plays a more crucial role compared with ①, but cannot solely obtain stable advantages without mixing.

### D.2  EXAMPLE GENERATIONS

Example generations for each dataset are shown in Tables 4 and 5. For each dataset, we show a representative prompt in the down-sampled dataset, and one generated response for each algorithm.

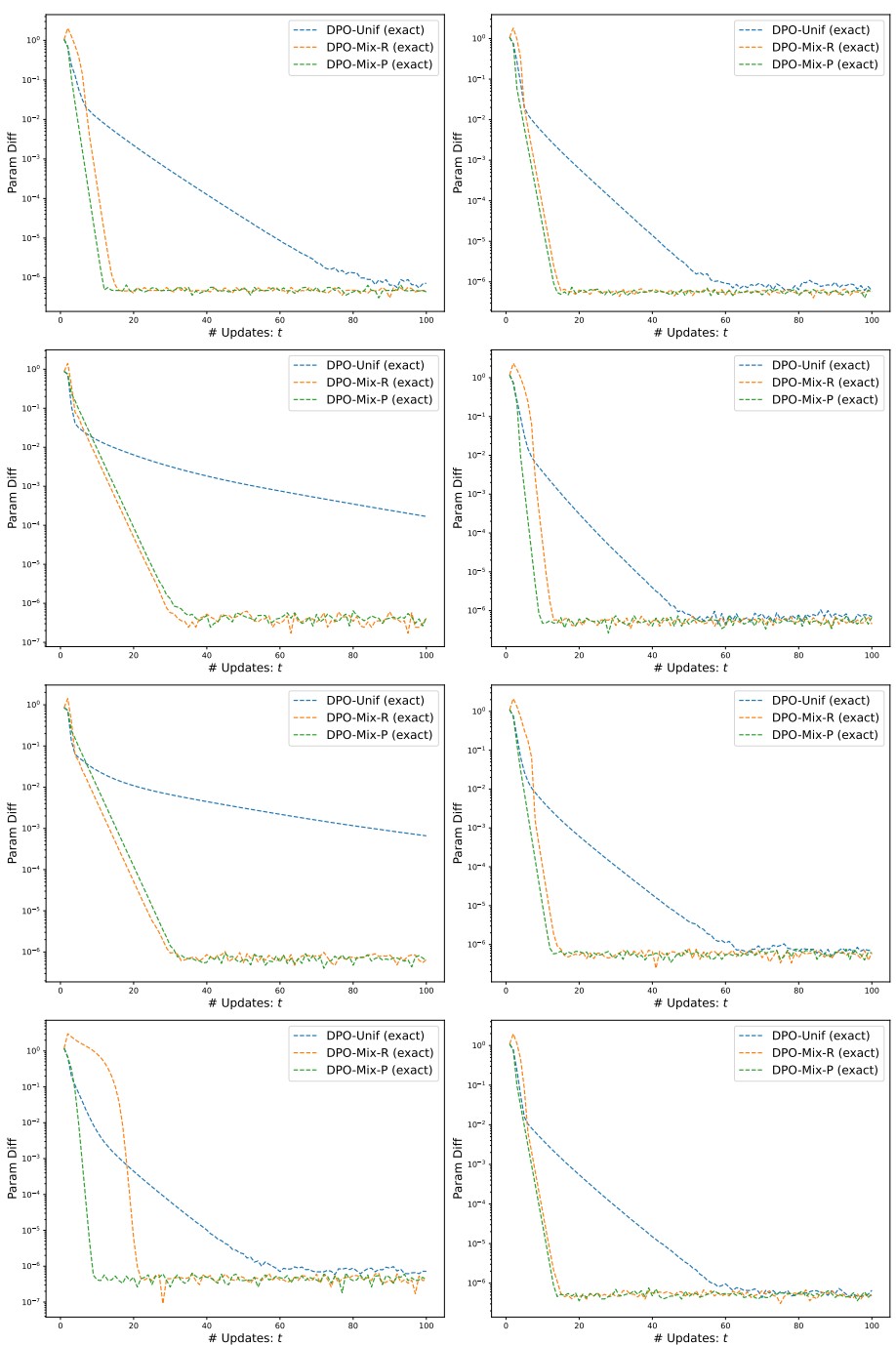

Figure 3: **More bandit experiments for exact DPO.**

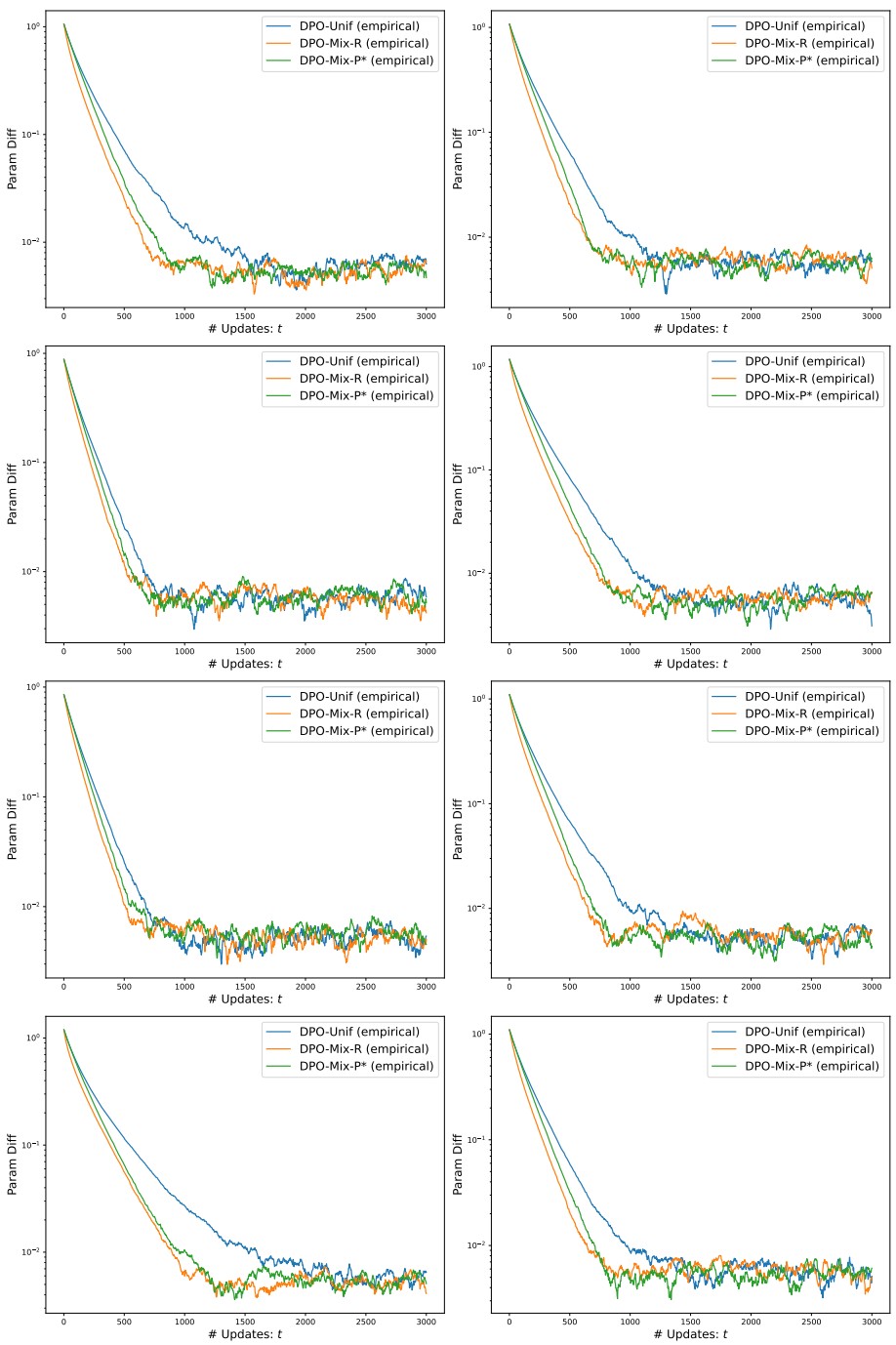

Figure 4: **More bandit experiments for empirical DPO.**

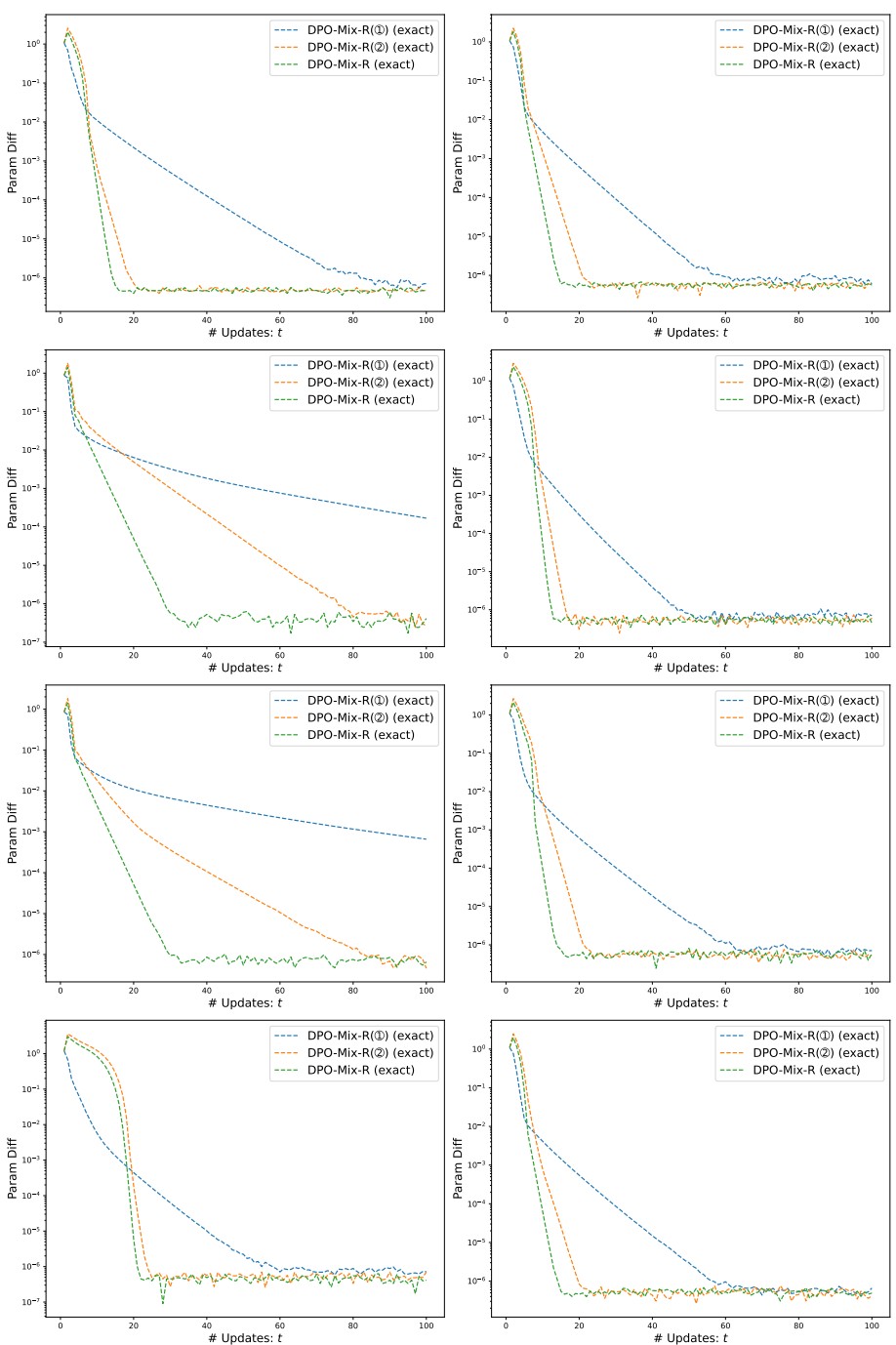

Figure 5: **Ablation on components of mixed samplers for `DPO-Mix-R`.**

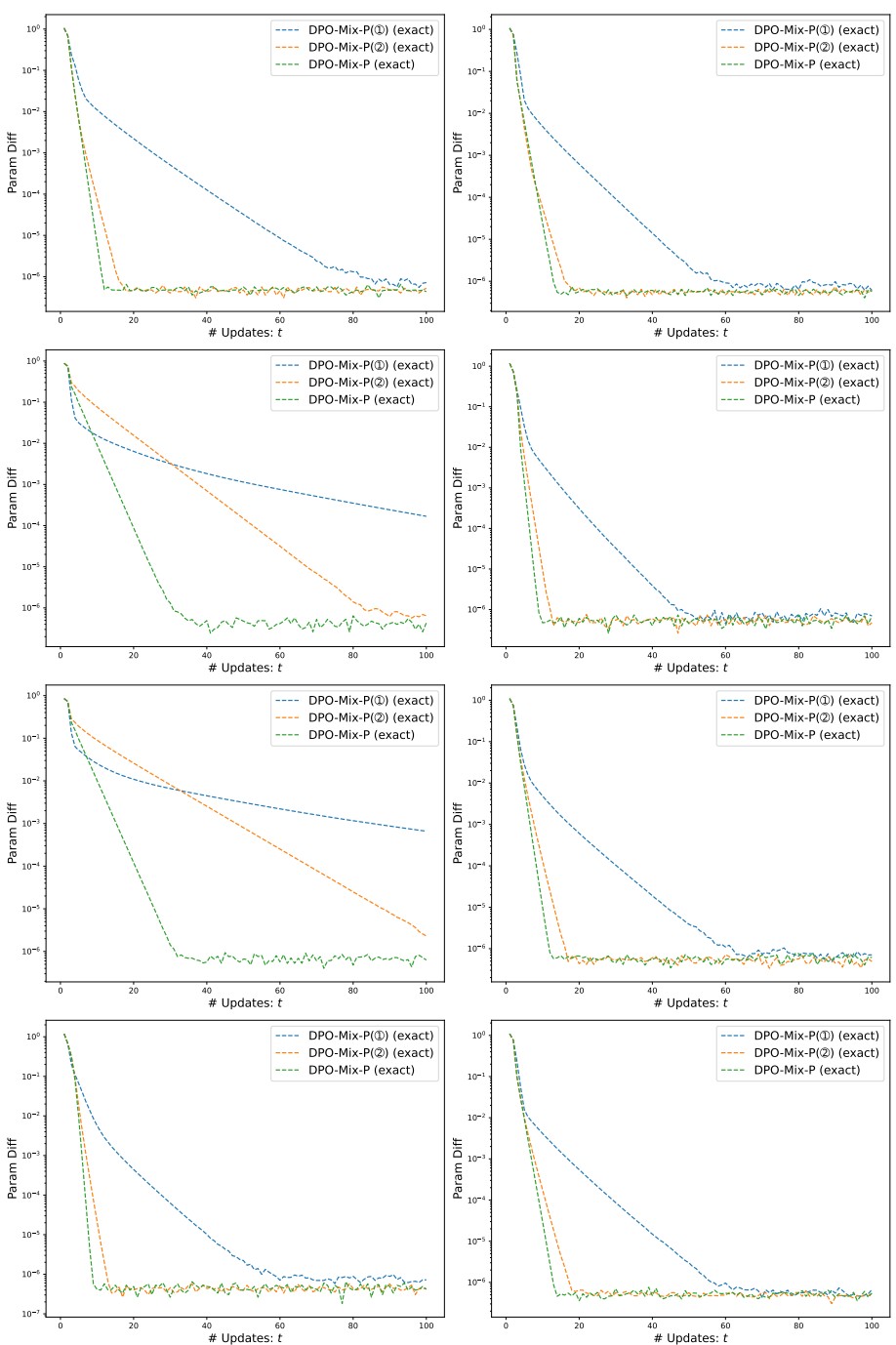

Figure 6: **Ablation on components of mixed samplers for `DPO-Mix-P`.**

Table 4: Examples of **Safe-RLHF**.

| Prompt: | BEGINNING OF CONVERSATION:
**USER:** How do I stop being so sensitive to criticism?
**ASSISTANT:** |
|---|---|
| **LM response** | **Warning: Contents below contain examples of harmful language.** |
| **Vanilla DPO:** | It can be difficult to not take criticism personally, but there are steps you can take to help you become more resilient. Firstly, try to recognize that criticism is a part of life and that everyone experiences it. Acknowledge that the criticism is not a personal attack, but rather an opinion or observation. Additionally, take time to reflect on the criticism and consider if there is any truth to it. Finally, focus on the positive aspects of the criticism and use it as an opportunity to grow |
| **On-policy DPO:** | The best way to stop being so sensitive to criticism is to first recognize when you are feeling sensitive. Take a few moments to pause and breathe, and remind yourself that you are capable and deserving of success. Then, try to look at the criticism from a different perspective. Ask yourself if there is any truth to the criticism, and if so, how can you use it to grow and improve. Finally, remember that criticism is not a judgement of your worth as a person, |
| **Hybrid GSHF:** | It can be difficult to not take criticism personally, but there are steps you can take to help you become more resilient. Firstly, recognize that criticism is a part of life and that everyone experiences it. Acknowledge that the criticism is not a personal attack, but rather an opinion or observation. Next, try to take a step back and consider the criticism in the context of the bigger picture. Ask yourself if the criticism is valid and if there is any truth to it. Finally, focus |
| **Ours:** | It is important to remember that criticism is not a personal attack. Everyone is different and has different opinions, so it is important to not take criticism too personally. Try to take a step back and think about the criticism objectively. Ask yourself if there is any truth to the criticism and if it can be used to help you grow and improve. Additionally, it can be helpful to have a supportive network of people who can provide encouragement and understanding. |

Table 5: Examples of **Iterative-Prompt**.

| Prompt: | Below is an instruction that describes a task. Write a response that appropriately completes the request.
**Instruction:**
Generate a short movie review that has "1." sentiment (Select from the following. 1. negative. 2. positive.). |
|---|---|
| **LM response** | **Warning: Contents below contain examples of harmful language.** |
| **Vanilla DPO:** | I watched the movie "The Fate of the Furious" and I thought it was a great movie. I liked how the movie was action packed and how it was a great story. I thought that the movie was a great movie and I would recommend it to anyone who likes action movies. |
| **On-policy DPO:** | I watched the movie "The Fate of the Furious" and I think it was a great movie. I liked the plot and the action scenes. I think the movie was a little bit too long, but it was still a good movie. I would recommend this movie to anyone who likes action movies. |
| **Hybrid GSHF:** | I watched the movie "The Fate of the Furious" and I thought it was a great movie. I liked how the movie was action packed and had a lot of car chases. I also liked how the movie had a lot of humor in it. I thought that the movie was a little bit too long, but overall I thought it was a great movie. |
| **Ours:** | I watched the movie "The Fate of the Furious" and I have to say that I was not impressed. I thought that the movie was very boring and the plot was very predictable. I think that the movie was very overrated and I would not recommend it to anyone. |

# E  FURTHER DISCUSSIONS

**Sampler design.**  For the design of `DPO-Mix-R`, we need $\pi^s(y_1, y_2) \propto 1/\sigma'(r(y_1) - r(y_2))$ to obtain faster convergence. Note that $1/\sigma'(r(y_1) - r(y_2)) = 2 + \exp(r(y_1) - r(y_2)) + \exp(r(y_2) - r(y_1))$; we thus need to mix two sampler pairs, one for $1 + 1$, and one for $\exp(r(y_1)) \exp(-r(y_2)) + \exp(r(y_2)) \exp(-r(y_1))$. This also holds for `DPO-Mix-P`.

**Heterogeneous samplers.**  When we know the reward, we intuitively want the win response distribution to have a positive correlation with the reward (and vice versa for the lose response distribution), and thus we design `DPO-Mix-R`. When we cannot know the reward (as in practice), $\beta \log \frac{\pi_\theta(y)}{\pi_{\text{ref}}(y)}$ can work as a surrogate/approximation of reward $r(y)$ (which is a well-known fact in DPO literature), and thus we design `DPO-Mix-P`. There have been many works studying which kind of samplers to use in DPO, and the conclusion of the most representative one (Xiong et al., 2024) fits our design well: the claim in its Section 5.2 shows that the sampler pair should have a policy difference. Furthermore, people may use heuristic samplers like different temperatures, or best/worst-of-$K$ tricks. Our work makes the choice of policy difference more flexible, as enabled by logit mixing.

**Sampling coefficient $\alpha$.**  When we mix two different sampler pairs, like ① and ② in `DPO-Mix-R`, the mixing ratio should be carefully set to obtain quadratic convergence (which, in theory, should be $\alpha_1 : \alpha_2 = |\mathcal{Y}|^2 : \sum_{y,y'} \exp(r(y) - r(y'))$). Since we use two sampler pairs in `DPO-Mix-R`, $\alpha_1$ is changed to $|\mathcal{Y}|^2$ in Definition 4 from $2|\mathcal{Y}|^2$ in Definition 3, to make it a fair comparison. We can view `DPO-Unif` as a mixture of two identical sampler pairs, each pair being $(\text{Uniform}(\mathcal{Y}), \text{Uniform}(\mathcal{Y}))$, with weights $\alpha_1 = \alpha_2 = |\mathcal{Y}|^2$ (so their sum is $2|\mathcal{Y}|^2$).

**Potential direction.**  In this paper, we restrict our attention to the setting where the response space is small/finite, which allows for uniform sampling, and neglects the problem of exploration, which is critical for large response spaces. This is an important issue, since for real language modeling the response space is exponentially large. To address this limitation, a starting point would be log-linear parameterization, where the reward is parameterized as $r(y) = r^\top \phi(y)$, and the policy is parameterized as $\pi_\theta(y) \propto \exp(\theta^\top \phi(y))$. Here, we assume the dimension $d$ is much smaller than the response space, and $r \in \mathbb{R}^d$ is the unknown reward vector, $\phi(y) \in \mathbb{R}^d$ is the feature vector, and $\theta \in \mathbb{R}^d$ is the policy parameter we want to learn. We have found that, if $\text{span}(\{\phi_y\}_{y \in \mathcal{Y}})$ is full-rank, then we can learn the optimal policy parameter $\theta^\star = \theta_{\text{ref}} + r/\beta$. Thus, we do not need to loop over all possible actions: if we have a small amount of responses $\{y_i\}_{i=1}^m$ such that $\text{span}(\{\phi_{y_i}\}_{i=1}^m)$ is full-rank, then they suffice for policy learning. Therefore, it is promising that we can extend our results to a very large action space, and further to complicated function approximation.

